# Generation of biophysical neuron model parameters from recorded electrophysiological responses

Jimin Kim[1], Minxian Peng[2], Shuqi Chen[2], Qiang Liu[2]*, Eli Shlizerman[1,3]*

[1]Department of Electrical and Computer Engineering, University of Washington, Seattle, United States; [2]Department of Neuroscience, City University of Hong Kong, Hong Kong, Hong Kong; [3]Department of Applied Mathematics, University of Washington, Seattle, United States

**\*For correspondence:**
qiangliuemail@gmail.com (QL);
shlizee@uw.edu (ES)

**Competing interest:** The authors declare that no competing interests exist.

## eLife Assessment

This study is a **valuable** contribution to the field of neuronal modeling by way of providing a method for rapidly obtaining neuronal physiology parameters from electrophysiological recordings. The method is **solid** as the generated models reproduce both ground-truth simulated data and empirical data, and there is now a quantitative comparison with other approaches.

**Abstract** Recent advances in connectomics, biophysics, and neuronal electrophysiology warrant modeling of neurons with further details in both network interaction and cellular dynamics. Such models may be referred to as ElectroPhysiome, as they incorporate the connectome and individual neuron electrophysiology to simulate neuronal activities. The nervous system of *Caenorhabditis elegans* is considered a viable framework for such ElectroPhysiome studies due to advances in connectomics of its somatic nervous system and electrophysiological recordings of neuron responses. In order to achieve a simulated ElectroPhysiome, the set of parameters involved in modeling individual neurons needs to be estimated from electrophysiological recordings. Here, we address this challenge by developing a deep generative estimation method called ElectroPhysiomeGAN (EP-GAN), which, once trained, can instantly generate parameters associated with the Hodgkin–Huxley neuron model (HH-model) for multiple neurons with graded potential response. The method combines generative adversarial network (GAN) architecture with recurrent neural network encoder and can generate an extensive number of parameters (>170) given the neuron's membrane potential responses and steady-state current profiles. We validate our method by estimating HH-model parameters for 200 simulated neurons with graded membrane potential followed by nine experimentally recorded neurons (where six of them are newly recorded) in the nervous system of *C. elegans*. Comparison of EP-GAN with existing estimation methods shows EP-GAN's advantage in the accuracy of estimated parameters and inference speed for both small and large numbers of parameters being inferred. In addition, the architecture of EP-GAN permits input with arbitrary clamping protocols, allowing inference of parameters even when partial membrane potential and steady-state currents profiles are given as inputs. EP-GAN is designed to leverage the generative capability of GAN to align with the dynamical structure of the HH-model and thus is able to achieve such performance.

## Introduction

Models of the nervous system aim to achieve biologically detailed simulations of large-scale neuronal activity through the incorporation of both structural connectomes (connectivity maps) and individual neural dynamics. The nervous system of *Caenorhabditis elegans* is considered a framework for such

a model as the connectome of its somatic nervous system for multiple types of interaction is mapped (*White et al., 1986*; *Varshney et al., 2011*; *Cook et al., 2019*). In addition to the connectome, advances in electrophysiological methodology allow the recording of whole-cell responses of individual neurons. These advances provide biophysically relevant details of individual neuro-dynamical properties and warrant a type of model for the *C. elegans* nervous system incorporating both the connectomes and individual biophysical processes of neurons. Such a model could be referred to as *ElectroPhysiome*, as it incorporates a layer of individual neural dynamics on top of the layer of intercellular interactions facilitated by the connectome.

The development of nervous system models that are further biophysically descriptive for each neuron, that is, modeling neurons using the Hodgkin–Huxley type equations (HH-model), requires fitting a large number of parameters associated with ion channels found in the system. For a typical single neuron, these parameters could be tuned via local optimizations of individual ion channel parameters estimated separately to fit their respective in vivo channel recordings such as activation/inactivation curves (*Hodgkin and Huxley, 1952*; *Willms, 2002*; *Willms et al., 1999*; *Nicoletti et al., 2019*; *Liu et al., 2018*; *Jiang et al., 2022*). Such a method requires multiple experiments to collect each channel data, and when such experiments are infeasible, the parameters are often estimated through hand-tuning. In the context of developing the ElectroPhysiome of *C. elegans*, the method would have to model approximately 300 neurons each including an order of hundreds of parameters associated with up to 15 to 20 ionic current terms (with some of them having unknown ion channel composition), which would require large experimental studies (*Nicoletti et al., 2019*). Furthermore, the fitted model may not be the unique solution as different HH-parameters can produce similar neuron activity (*Marder and Goaillard, 2006*; *Marder and Taylor, 2011*; *Prinz et al., 2003*; *Prinz et al., 2004*). As these limitations also apply for general neuron modeling tasks beyond *C. elegans* neurons, there has been an increasing search for alternative fitting methods requiring less experimental data and manual interventions.

A promising direction in associating model parameters with neurons has been the simultaneous estimation of all parameters of an individual neuron given only electrophysiological responses of cells, such as membrane potential responses and steady-state current profiles. Such an approach requires significantly less experimental data per neuron and offers more flexibility with respect to trainable parameters. The primary aim of this approach is to model macroscopic cell behaviors in an automated fashion. Indeed, several methods adopting the approach have been introduced. *Buhry et al., 2012* and *Laredo et al., 2022* utilized the differential evolution (DE) method to simultaneously estimate the parameters of a 3-channel HH-model given the whole-cell membrane potential responses recording (*Buhry et al., 2012*; *Laredo et al., 2022*). *Naudin et al., 2022b* further developed the DE approach and introduced the multi-objective differential evolution (DEMO) method to estimate 22 HH-parameters of three non-spiking neurons in *C. elegans* given their whole-cell membrane potential responses and steady-state current profiles (*Naudin et al., 2022b*). The study was a significant step toward modeling whole-cell behaviors of *C. elegans* neurons in a systematic manner. From a statistical standpoint, *Wang et al., 2022* used the Markov Chain–Monte Carlo method to obtain the posterior distribution of channel parameters for HH-models featuring three and eight ion channels (two and nine parameters, respectively) given the simulated membrane potential responses data (*Wang et al., 2022*). From an analytic standpoint, Valle et al. 2022 suggested an iterative gradient descent-based method that directly manipulates the HH-model to infer three conductance parameters and three exponents of activation functions given the measurements of membrane potential responses (*Valle and Madureira, 2022*). Recent advances in machine learning gave rise to deep learning-based methods which infer steady-state activation functions and posterior distributions of three-channel HH-model parameters inferred by an artificial neural network model given the membrane potential responses data (*Gonçalves et al., 2020*; *Estienne, 2021*).

While these methods suggest that simultaneous parameter estimation from macroscopic cell data is indeed possible through a variety of techniques, it is largely unclear whether they can be extended to fit more detailed HH-models featuring a large number of channels and parameters (e.g., *C. elegans* neurons) (*Nicoletti et al., 2019*). Furthermore, for most of the above methods, the algorithms require an independent (from scratch) optimization process for fitting each individual neuron, making it difficult to scale up the task toward a large number of neurons.

**Figure 1.** Estimation of HH-model parameters from membrane potential and steady-state current profiles. Given the membrane potential responses (V) and steady-state current profiles (IV) of a neuron, the task is to predict biophysical parameters of the Hodgkin–Huxley-type neuron model (left). We use the Encoder-Generator approach to predict the parameters (right).

Here, we propose a new machine learning approach that aims to address these aspects for the class of non-spiking neurons, which constitute the majority of neurons in *C. elegans* nervous system (*Goodman et al., 1998*). Specifically, we develop a deep generative neural network model (GAN) combined with a recurrent neural network (RNN) Encoder called ElectroPhysiomeGAN (EP-GAN), which directly maps electrophysiological recordings of a neuron, for example, membrane potential responses and steady-state current profiles, to HH-model parameters of arbitrary dimensions (*Figure 1*). EP-GAN can be trained with simulation data informed by a generic HH-model encompassing a large set of arbitrary ionic current terms and thus can generalize its modeling capability to multiple neurons. Unlike typical GAN architecture trained solely with adversarial losses, we propose to implement an additional regression loss for reconstructing the given membrane potential responses and current profiles from generated parameters, thus improving the accuracy of the generative model. In addition, due to the RNN component of EP-GAN, the approach supports input data with missing features such as incomplete membrane potential responses and current profiles.

We validate our method to estimate HH-model parameters of 200 simulated non-spiking neurons followed by applying it to three previously recorded non-spiking neurons of *C. elegans*, namely RIM, AFD, and AIY. Studies have shown that membrane potential responses of these neurons can be well modeled with typical HH-model formulations with 22 parameters (*Naudin et al., 2022b*; *Naudin et al., 2021*). We show that when trained with a more detailed HH-model consisting of 16 ionic current terms resulting in 175 trainable parameters, EP-GAN can predict parameters reproducing their membrane potential responses with higher accuracy in the reconstruction of membrane potential with significantly faster inference speed than existing algorithms such as differential evolution and genetic algorithms. Through ablation studies on input data, we show that EP-GAN retains its prediction capability when provided with incomplete membrane potential responses and steady-state current profiles. We also perform ablation studies on EP-GAN architecture components to elucidate each component's contributions toward the accuracy of the predicted parameters. To further test EP-GAN, we estimate HH-model parameters for six newly recorded non-spiking *C. elegans* neurons: AWB, AWC, URX, RIS, DVC, and HSN, whose membrane potential responses were not previously modeled.

Our results suggest that EP-GAN can learn a translation from electrophysiologically recorded responses and propose projections of them to parameter space. EP-GAN method is currently limited to non-spiking neurons in *C. elegans* as it was designed and trained with the HH-model describing the ion channels of these neurons. EP-GAN applications can potentially be extended toward resolving neuron parameters in other organisms since non-spiking neurons are found within animals across different species (*Koch et al., 2006*; *Roberts and Bush, 1981*; *Davis and Stretton, 1989a*; *Davis and Stretton, 1989b*; *Burrows et al., 1988*; *Laurent and Burrows, 1989a*; *Laurent and Burrows, 1989b*; *Naudin, 2022a*).

## Results

We evaluate EP-GAN with respect to four existing evolutionary algorithms introduced for general parameter estimation: NSGA2, DEMO, GDE3, and NSDE. Specifically, NSGA2 is a variant of the Genetic Algorithm (GA) that uses a non-dominated sorting survival strategy and is a commonly used benchmark algorithm for multi-objective optimization problems that include HH-model fitting (*Deb et al., 2000*; *Hay et al., 2011*; *van Geit et al., 2008*). DEMO, GDE3, and NSDE are variants

of Differential Evolution (DE) algorithms that combine DE mutation with Pareto-based ranking and crowding distance sorting applied in NSGA2's survival strategy (*Robič and Filipič, 2005*; *Kukkonen and Lampinen, 2005*; *Angira and Babu, 2005*). These methods have been proposed as more effective methods than direct DE for the estimation of HH-model parameters (*Naudin et al., 2022b*; *Rumbell et al., 2016*; *Octeau et al., 2019*; *Buhry et al., 2009*). In particular, DEMO has been successfully applied to estimate HH-model parameters for non-spiking neurons in *C. elegans* (*Naudin et al., 2022b*). All four methods support multi-objective optimization over the large parameter space, allowing them to have similar setups as EP-GAN. All four methods were implemented in Python where DEMO uses the algorithm proposed in *Naudin et al., 2022b* whereas NSGA2, GDE3, and NSDE were implemented using Pymoo package (*Blank and Deb, 2020*).

For the HH-model to be estimated, we use the formulation introduced in *Nicoletti et al., 2019*; *Nicoletti et al., 2024*. The model features 16 ion channels that were found in *C. elegans* and other organisms expressing homologous channels and is considered the most detailed neuron model for the organism (see 'Ionic currents modeling' for the mathematical description of channels). The model has a total of 203 parameters, of which we identify 175 of them have the approximate ranges with lower and upper bounds that can be inferred from the literature (*Liu et al., 2018*; *Naudin et al., 2021*; *Izhikevich, 2007*). We thus target these 175 parameters as trainable parameters for all methods. For a detailed list of all 203 parameters included in the HH-model and 175 parameters used for training, see *Nicoletti et al., 2024* and the included table *predicted parameters* in *Supplementary file 1*.

## Predictions on simulated neurons

We first validate EP-GAN by training and testing using simulated neurons. Each simulated neuron training sample consists of two inputs: (i) simulated membrane potential traces concatenated with associated external stimuli traces and (ii) steady-state currents across 18 voltage points. For each neuron, the output is the set of 175 HH-parameters to be inferred. Each membrane potential trace is

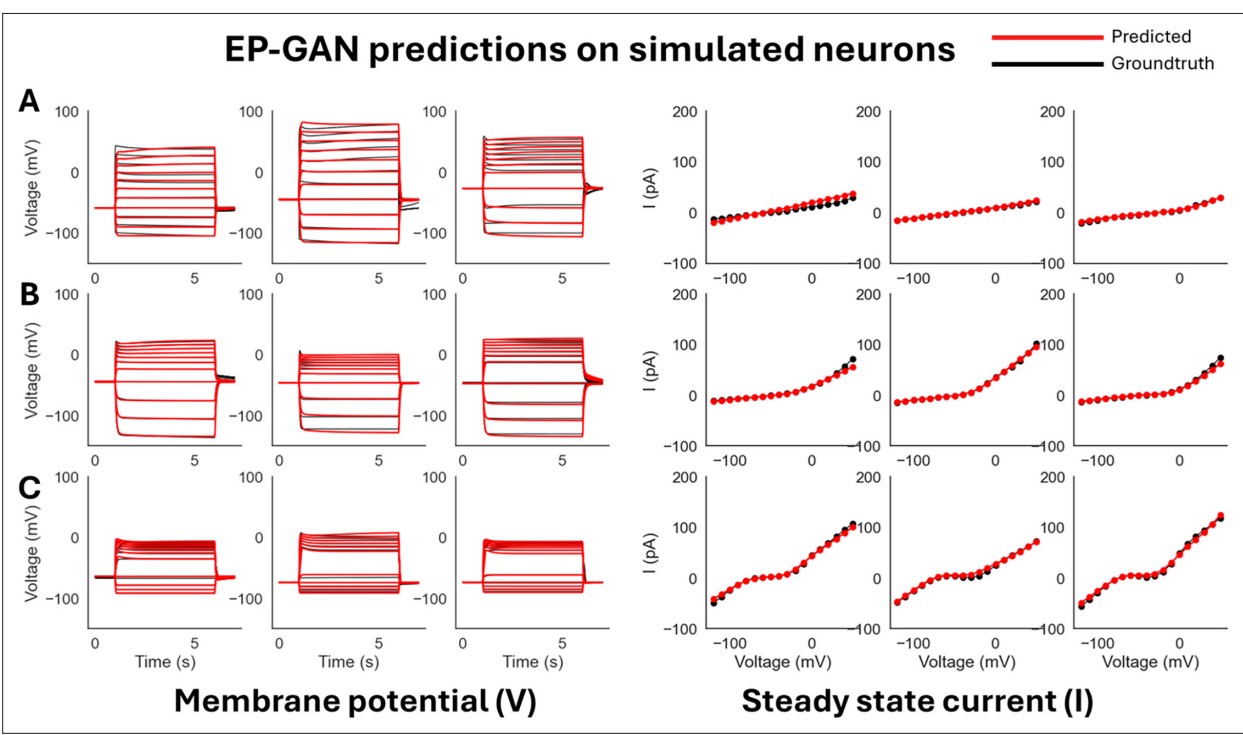

**Figure 2.** ElectroPhysiomeGAN (EP-GAN) (32k) predictions on simulated neurons. (**A**) EP-GAN predicted membrane potential traces and steady-state currents (red) overlaid on top of groundtruth counterparts (black) for transient outward rectifier neuron type. (**B**) Outward rectifier neuron type. (**C**) Bistable neuron type.

The online version of this article includes the following figure supplement(s) for figure 2:

**Figure supplement 1.** Root mean square error (RMSE) error distribution (averaged over pre-, mid-, post-activation time periods) for the simulated neurons ($n = 200$) in test set.

simulated for 15 s for a given stimulus according to the current-clamp protocol where the stimulation is applied for 5 s at [5, 10] s and no stimulation is applied at $t = [0, 5)$ (pre-activation) and $t = (10, 15]$ (post-activation). These time intervals are chosen to ensure sufficient stabilization periods before and after stimulation. For the membrane potential input, the responses during $t = [4, 11]$ interval are used consistent with the time interval used by experimental recordings. Similarly, steady-state currents are computed across 18 voltage states according to voltage-clamp protocol (see Table 4 for detailed current/voltage clamp protocols used for simulated neurons). The output HH-parameters are of the simulated neurons chosen randomly from lower and upper bounds as previously described. For training EP-GAN, we simulate a total of 32,000 (32k) neurons where EP-GAN achieves both good predictive performance and training time. Specifically, 32k is a training data size in which membrane potential errors from the test set ($N = 200$) are within the average root mean square error (RMSE) recording error (4.8 mV, averaged over pre-, mid-, post-activation periods) obtained from experimental neurons with multiple membrane potential recording data. For more details on generating simulated neuron training samples, see 'Generating training data'.

To initially test EP-GAN performance, we evaluate EP-GAN predicted parameters for 200 simulated neurons outside of the training set (test set). To emulate *C. elegans*' neuronal diversity, neurons in the test set are divided into three response types: (i) transient outward rectifier type, (ii) outward rectifier type, and (iii) bistable type – that are currently found in non-spiking neurons of *C. elegans* according to their steady-state responses (*Figure 2*; *Liu et al., 2018*; *Naudin et al., 2022b*). For a given neuron being evaluated, EP-GAN predicted HH-parameters are obtained as follows: for each training epoch, EP-GAN generates a set of HH-parameters for the neuron and at the end of the training, the parameter set which achieved the lowest RMSE of membrane potential responses averaged across three time intervals – pre-activation [4, 5) s, mid-activation [5, 10] s, and post-activation (10, 11] s – with respect to ground truths is reported (detailed descriptions of its calculation provided in Appendix 1 andFigure 5A). In the case of multiple $N$-neurons being evaluated, the same procedure is followed except EP-GAN generates $N$-parameter sets in parallel at each epoch. Such multi-neuron inference is possible due to EP-GAN being a neural network, where parallel processing of inputs can be done with minimal impact on inference speed. Using these procedures, EP-GAN predicted HH-parameters result in mean membrane potential RMSE error of 2.37 mV for the test set (see *Figure 2* for representative samples and *Appendix 1—table 1* for the detailed breakdown of the errors). These errors are within the mean recording RMSE error of 4.8 mV obtained from experimental neurons, and their distributions were skewed unimodal type, where the majority of the errors fall within 4 mV (*Figure 2—figure supplement 1*).

## Predictions on experimental neurons

We apply EP-GAN trained and tested on simulated data to predict HH-parameters for nine experimentally recorded non-spiking neurons found in *C. elegans*: RIM, AFD, AIY, AWB, AWC, URX, RIS, DVC, and HSN. Among these neurons, AWB, AWC, URX, RIS, DVC, and HSN are novel recording data and were not previously modeled, whereas RIM, AIY, and AFD neurons are publicly available and their modeling descriptions were elaborated by previous works (*Naudin et al., 2022b*; *Naudin et al., 2021*; *Naudin et al., 2022c*). Similar to the prediction scenario on simulated neurons, we categorize experimental neurons into different response types according to their steady-state current responses. In particular, we classify (RIM, DVC, HSN) as transient outward rectifier type, (AIY, URX, RIS) as outward rectifier type, and (AFD, AWB, AWC) as bistable type. For all experimental neurons, simulation protocols outlined in Table 4 are used to generate membrane potential and steady-state responses of predicted parameters.

We compare the performance of EP-GAN with four existing parameter inference methods: NSGA2, DEMO, GDE3, and NSDE. Unlike EP-GAN, which can optimize multiple neurons in parallel, these are evolutionary methods where the optimization is done **from scratch** for each neuron. We therefore evaluate their respective performances relative to EP-GAN by normalizing the *total number of simulated neuron samples* during the entire optimization task. Specifically, for all methods, we set the maximum number of neuron samples used during optimization to equal sizes. For example if EP-GAN is trained with 32k neuron samples to predict nine neurons, NSGA2, DEMO, GDE3, and NSDE are each allocated up to 3.5k samples during the search phase of HH-parameters for each neuron, thus adding up to a total of 32k samples for all nine neurons. To test how the performance of each method

scales with the amount of samples during optimization, we evaluate each method with both 32k and 64k total neuron samples.

The parameter selection process for NSGA2, DEMO, GDE3, and NSDE is as follows: during the search phase for each neuron, the parameter set candidates (i.e., population) are recorded at each iteration. At the end of the search phase, the steady-state current profile of each parameter set candidate is evaluated with respect to the experimentally known bifurcation structure (i.e., *dI/dV*) of the neuron being inferred (e.g., bistable type). Upon evaluation, only the parameter sets satisfying the *dI/dV* bound constraints (∼ 98% confidence interval) are kept. Such an initial selection process is similar to the ones employed by previous methods utilizing evolutionary algorithms (*Naudin et al., 2022b*). The *dI/dV* bound constraints are also used for generating EP-GAN training data (see 'Generating training data' for more detail). The final parameter set is then chosen by selecting the one with the lowest RMSE membrane potential responses error averaged across pre-, mid-, and post-activation periods identical to that of the EP-GAN parameter selection process. For DE methods (DEMO, GDE3, and NSDE), we follow the same configurations used in the literature to set their optimization scheme (*Naudin et al., 2022b*). Specifically, we set the crossover parameter CR and scale factor F to 0.3 and 1.5, respectively. For all four methods, NP (population size) is set to 600 with a total of 6 and 12 iterations (i.e., ∼ 3.6*k* and ∼ 6.2*k* samples per neuron for 32k and 64k total neuron samples, respectively). For all methods, loss functions identical to the ones used for EP-GAN training (mean absolute errors, see 'Materials and methods' for detail) are used to calculate the errors for membrane potential responses and steady-state currents for multi-objective optimization.

### Small HH-model scenarios (47 parameters)

We first test EP-GAN and existing methods with a 'smaller' version of the HH-model of 47 parameters where the individual channel parameters (*n* = 129) are frozen to default values given by *Nicoletti et al., 2024*. The considered parameters consist of 16 conductance parameters of each channel

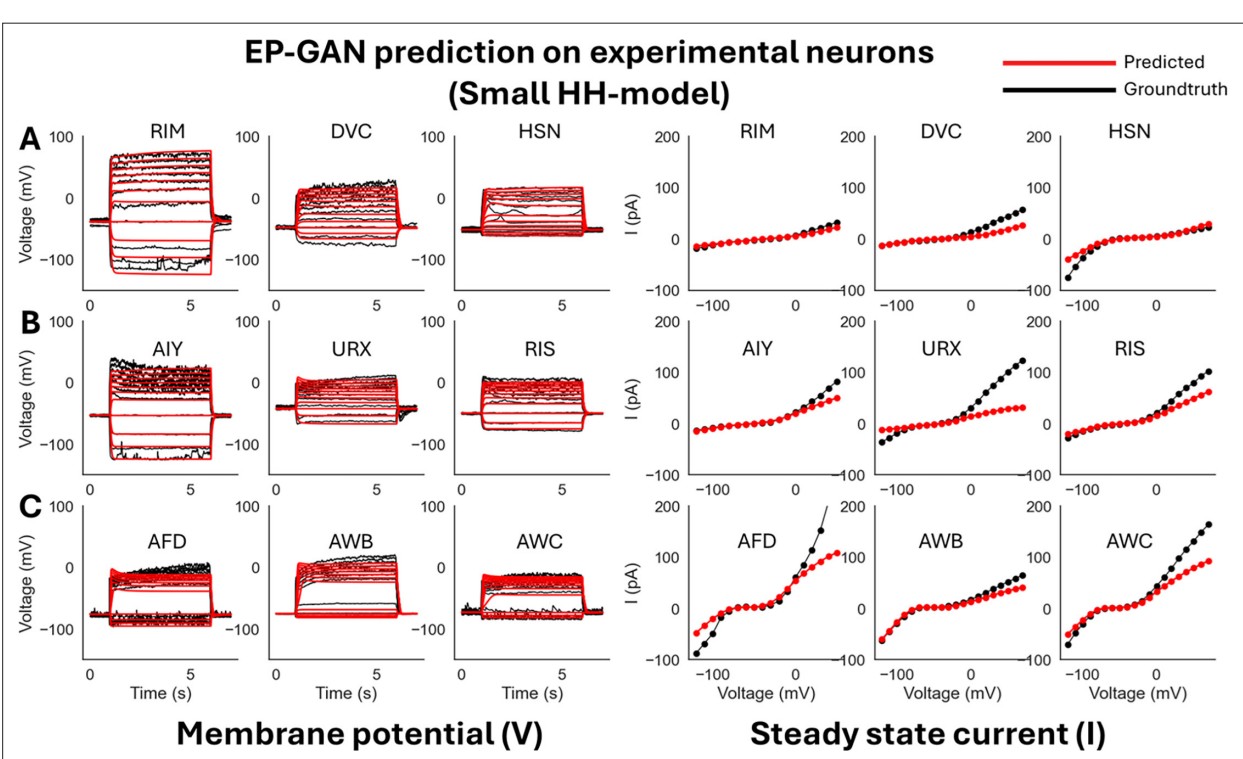

**Figure 3.** ElectroPhysiomeGAN (EP-GAN) (32k) prediction on experimental neurons (small HH-model). (**A**) EP-GAN predicted membrane potential traces and steady-state currents (red) overlaid on top of groundtruth counterparts (black) for transient outward rectifier neuron type (RIM, DVC, HSN). (**B**) Outward rectifier neuron type (AIY, URX, RIS). (**C**) Bistable neuron type (AFD, AWB, AWC).

The online version of this article includes the following figure supplement(s) for figure 3:

**Figure supplement 1.** Small HH-model GDE3, NSDE, DEMO, NSGA2 predictions (sample size = 32k) on experimental neurons.

**Table 1.** Small HH-model scenarios root mean square error (RMSE) errors for predicted membrane potential responses and steady-state currents.

Each method is tested with 32k or 64k total sample sizes, where the top row shows membrane potential responses RMSE errors averaged across pre-activation, mid-activation, post-activation periods, and the bottom row shows steady-state currents RMSE errors across 18 voltage values. The lowest membrane potential responses RMSE error is marked bold for each neuron.

| Method | Sample size | Median error | RIM | DVC | HSN | AIY | URX | RIS | AFD | AWB | AWC |
|---|---|---|---|---|---|---|---|---|---|---|---|
| | 32k | 16.9 mV | 20.9 | 58.1 | 8.9 | 13.0 | 16.9 | 25.4 | 6.3 | 32.5 | 4.7 |
| | | 5.9 pA | 5.8 | 5.8 | 19.8 | 4.8 | 2.4 | 5.9 | 19.1 | 7.2 | 11.9 |
| GDE3 | 64k | 6.0 mV | 11.5 | 24.1 | 13.7 | 6.0 | 5.1 | 7.1 | 4.1 | 5.6 | 3.6 |
| | | 7.9 pA | 7.5 | 4.6 | 6.8 | 7.2 | 18.7 | 7.9 | 42.1 | 17.6 | 46.7 |
| | 32k | 7.1 mV | 38.7 | 8.3 | 20.2 | 5.7 | 7.1 | 11.0 | 5.5 | 6.3 | 6.6 |
| | | 13.6 pA | 3.1 | 21.5 | 9.7 | 13.6 | 14.2 | 5.7 | 24.7 | 9.6 | 14.5 |
| NSDE | 64k | 5.5 mV | 15.4 | 8.7 | 20.2 | 13.5 | 5.2 | 5.5 | 4.9 | 4.9 | 4.0 |
| | | 9.7 pA | 2.6 | 5.9 | 9.7 | 3.4 | 11.7 | 3.2 | 64.4 | 12.4 | 19.0 |
| | 32k | 6.7 mV | 35.9 | 14.1 | 5.8 | 13.2 | 9.0 | 6.7 | 5.0 | 4.9 | 3.8 |
| | | 11.5 pA | 6.5 | 13.8 | 14.6 | 5.2 | 5.4 | 9.7 | 23.8 | 11.5 | 18.7 |
| DEMO | 64k | 4.8 mV | 12.3 | 10.5 | 5.8 | 10.2 | 4.8 | 4.7 | **3.1** | 4.4 | 2.9 |
| | | 14.6 pA | 4.4 | 6.5 | 14.6 | 4.1 | 15.2 | 10.3 | 41.0 | 23.1 | 17.8 |
| | 32k | 7.5 mV | 12.4 | 15.4 | 2.6 | 9.8 | 6.1 | 6.4 | 7.5 | 9.4 | 6.6 |
| | | 10 pA | 4.0 | 7.4 | 14.3 | 4.6 | 16.6 | 5.0 | 14.4 | 11.9 | 10.0 |
| NSGA2 | 64k | 4.3 mV | 10.5 | 19.0 | 5.2 | 13.3 | 3.5 | 3.2 | 4.2 | 4.3 | 3.1 |
| | | 8.4 pA | 8.4 | 1.8 | 5.2 | 2.3 | 21.2 | 6.5 | 26.6 | 12.6 | 49.4 |
| | 32k | 2.5 mV | 3.4 | 2.4 | 1.6 | 2.5 | 3.2 | 1.7 | 4.9 | 2.5 | **2.0** |
| | | 13.8 pA | 4.0 | 13.8 | 10.3 | 10.7 | 38.9 | 16.8 | 48.0 | 9.6 | 28.9 |
| EP-GAN (ours) | 64k | **2.4 mV** | **3.4** | **2.4** | **1.6** | **2.4** | **2.9** | **1.4** | 3.4 | **2.5** | 2.1 |
| | | 13.1 pA | 3.2 | 13.9 | 2.5 | 9.8 | 16.5 | 13.1 | 49.7 | 11.6 | 27.9 |

($g_{Ch}$), 4 reversal potentials for calcium, potassium, sodium, and leak channels ($V_{Ca}, V_K, V_{Na}, V_L$), 1 cell capacitance $C$, and 26 initial conditions for membrane potential $V_0$ and channel activation variables ($m_0, h_0$). Such a parameter set is commonly targeted when fitting HH-models for individual neurons (*Nicoletti et al., 2019*; *Nicoletti et al., 2024*). For all methods, we test both 32k and 64k total sample sizes for the inference of nine experimental neurons. *Figure 3* illustrates that EP-GAN can reconstruct membrane potential responses close to ground truth responses. Indeed, upon inspecting the RMSE error for membrane potential responses (averaged over pre-, mid-, and post-activation), EP-GAN (32k) median error of 2.5 mV over nine neurons is ∼ 50% lower than that of NSGA2 (64k) 4.3 mV followed by DEMO (64k) 4.8 mV, NSDE (64k) 5.5 mV, and GDE3 (64k) 6.0 mV (*Table 1*, *Appendix 1—table 2*, Figure 5B).

Among all nine neurons being inferred, EP-GAN (32k) showed the best accuracy for HSN with 1.6 mV and the lowest accuracy for AFD with 4.9 mV. With EP-GAN (64k), the median error further decreased (2.5mV → 2.4mV), where URX and RIS errors improved by 0.3 mV and AFD error improved by 1.5 mV over their 32k counterparts. Interestingly, we note that the high accuracy of EP-GAN in predicting membrane potential is not necessarily complemented with steady-state currents (*Table 1*, Figure 5B). EP-GAN's overall steady-state current errors are generally higher than those of existing methods. A possible reason could be that the majority of these errors stem from lower and upper voltage ranges where the recording variations are high, thus potentially causing a conflict with membrane potential responses optimization. Such a competitive nature between membrane potential vs. steady-state current optimizations has indeed been reported in previous works (*Naudin et al., 2022b*).

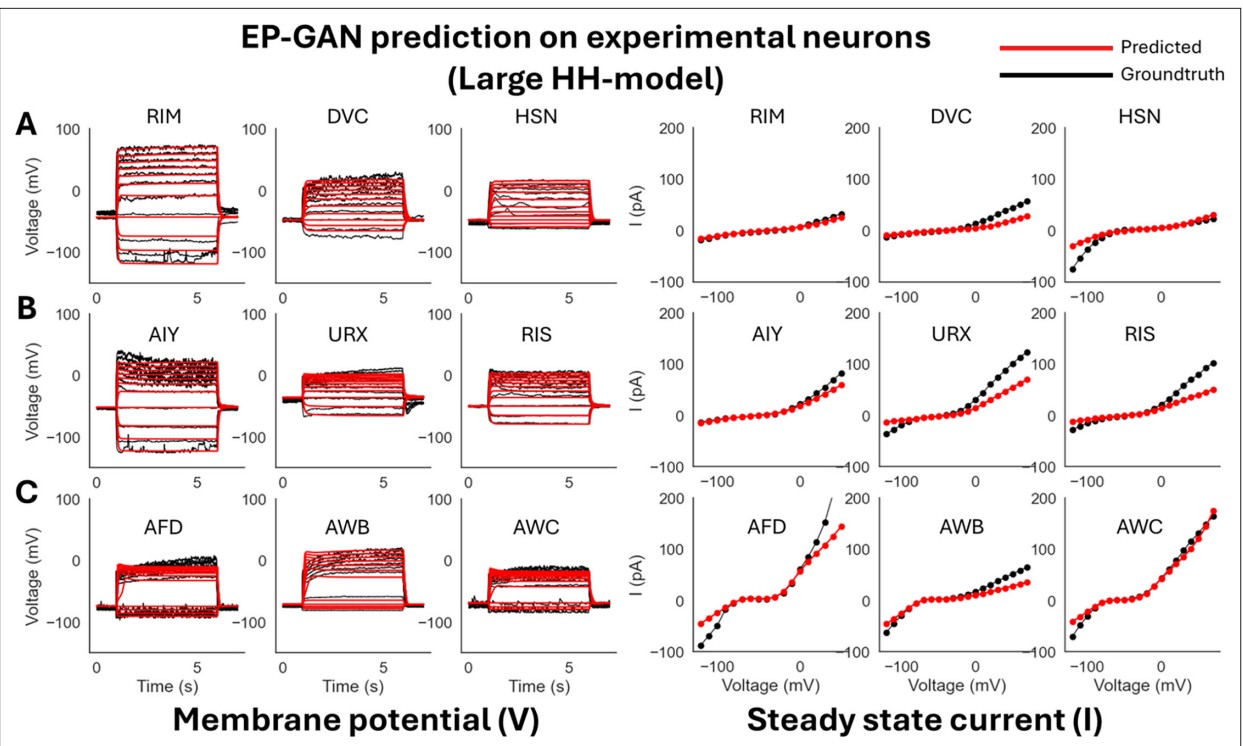

**Figure 4.** ElectroPhysiomeGAN (EP-GAN) (32k) prediction on experimental neurons (large HH-model). (**A**) EP-GAN predicted membrane potential traces and steady-state currents (red) overlaid on top of groundtruth counterparts (black) for transient outward rectifier neuron type (RIM, DVC, HSN). (**B**) Outward rectifier neuron type (AIY, URX, RIS). (**C**) Bistable neuron type (AFD, AWB, AWC).

The online version of this article includes the following figure supplement(s) for figure 4:

**Figure supplement 1.** Large HH-model GDE3, NSDE, DEMO, NSGA2 predictions (sample size = 32k) on experimental neurons.

### Large HH-model scenarios (175 parameters)

We expand the domain of parameters being inferred by testing with respect to all 175 HH-model's trainable parameters including the 47 parameters from the previous scenario + 129 channel parameters. The inclusion of channel parameters allows methods to further fine-tune the HH-model. The minimum and maximum values for channel parameters are set to ±50% from their default values (see 'Generating training data' for more detail). Such a task introduces further challenges as the methods need to estimate ×3 more parameters compared to the small HH-model scenario. From *Figures 4 and 5B*, *Table 2*, and *Appendix 1—table 3*, we see that while EP-GAN (32k) median membrane potential error increases slightly from 2.5 mV to 2.7 mV, its performance gaps over existing methods widen with ∼ 60% lower error than NSGA2 (64k) 7.5 mV, followed by NSDE (64k) 8.6 mV and DEMO (64k), GDE3 (64k) 10.5 mV. EP-GAN trained for large HH-model also slightly improved overall steady-state current error (13.8pA → 12.4pA) alongside membrane potential errors for RIM (3.4mV → 3.2mV) and URX (3.2mV → 3.0mV), indicating different selectivity for individual neurons for small vs. large HH-model. Further increasing the sample size to 64k improved median errors of both membrane potential (2.7mV → 2.6mV) and steady-states responses (12.4pA → 8.6pA). Taken together, these results show EP-GAN's predicting capabilities for HH-parameters with higher membrane potential responses accuracy and its ability to generalize to a larger parameter space with minimal performance loss.

### Ablation studies

EP-GAN architecture also allows its membrane potential inputs to have arbitrary current-clamp protocol due to its RNN encoder component. To test the robustness of EP-GAN when incomplete input data is given, we provide the model with membrane potential responses and steady-state current inputs with missing data points. For each membrane potential responses and current profile, the data is reduced by 25%, 50%, and 75% each. For membrane potential responses data, the ablation is done on stimulus space where a 50% reduction corresponds to removing the upper half of the

membrane potential response traces each associated with a stimulus. For the steady-state current profile, we remove the first *n*-data points where they are instead extrapolated using linear interpolation with existing data points.

Our results show that EP-GAN largely preserves accuracy even when both membrane potential and steady-state current inputs are masked (*Figure 6*, *Figure 6—figure supplement 1* [predicted steady-state currents], *Table 3*). In particular, EP-GAN preserves median membrane potential error (3.3mV) up to 50% remaining in its inputs but becomes less accurate when up to 25% input remains (3.3mV → 5.4mV). Surprisingly, AFD neuron membrane potential error is improved when only 50% of input data is considered (5.2mV → 4.5mV). These results could be attributed to the random input masking employed during EP-GAN training (see 'Materials and methods' for detail), which allows EP-GAN to make robust predictions even when conditioned with varying degrees of masked inputs.

We also perform ablation studies on EP-GAN architecture by removing each loss component of the Generator module, allowing us to evaluate their relative contributions to accuracy. For all loss ablation scenarios, simulation protocols outlined in *Table 4* are used to generate membrane potential and steady-state responses of predicted parameters. From *Table 3* bottom, we see that removing the membrane potential loss term (V) results in a loss in performance for RIM and AIY but increases in accuracy for AFD. The result is consistent with input data ablation scenarios indicating AFD's higher dependence on steady-state responses for good EP-GAN prediction. Upon removing the steady-state current reconstruction loss term (IV) in addition to the membrane potential reconstruction loss, we see a further reduction in overall performance. These results highlight the significance of the reconstruction losses in aligning the Generator to produce the desired outputs (*Table 3*).

**Table 2.** Large HH-model scenarios root mean square error (RMSE) errors for predicted membrane potential responses and steady-state currents.

Each method is tested with 32k or 64k total sample sizes, where the top row shows membrane potential responses RMSE errors averaged across pre-activation, mid-activation, post-activation periods, and the bottom row shows steady-state currents RMSE errors across 18 voltage values. The lowest membrane potential responses RMSE error is marked bold for each neuron.

| Method | Sample size | Median error | RIM | DVC | HSN | AIY | URX | RIS | AFD | AWB | AWC |
|---|---|---|---|---|---|---|---|---|---|---|---|
| | | 12.8 mV | 14.0 | 12.5 | 12.8 | 19.4 | 9.0 | 15.2 | 29.4 | 10.8 | 9.2 |
| | 32k | 9.6 pA | 3.2 | 23.5 | 12.8 | 10.2 | 6.3 | 6.0 | 7.9 | 18.1 | 9.6 |
| | | 10.5 mV | 14.0 | 11.0 | 10.5 | 11.7 | 12.4 | 9.0 | 5.0 | 9.5 | 4.7 |
| GDE3 | 64k | 4.9 pA | 3.2 | 6.5 | 7.4 | 3.8 | 4.8 | 4.0 | 16.9 | 14.8 | 4.9 |
| | | 16.1 mV | 31.5 | 19.0 | 8.7 | 12.1 | 8.6 | 23.8 | 9.8 | 27.1 | 16.1 |
| | 32k | 7.2 pA | 7.8 | 8.0 | 9.6 | 6.8 | 5.1 | 4.2 | 18.2 | 7.2 | 6.2 |
| | | 8.6 mV | 33.9 | 8.6 | 12.4 | 5.9 | 8.6 | 8.0 | 16.6 | 12.5 | 4.6 |
| NSDE | 64k | 8.1 pA | 4.0 | 29.6 | 8.1 | 4.7 | 5.1 | 4.3 | 9.1 | 15.0 | 14.9 |
| | | 16.6 mV | 28.0 | 17.8 | 16.6 | 10.2 | 22.9 | 18.1 | 6.0 | 13.1 | 5.1 |
| | 32k | 11.9 pA | 7.6 | 11.9 | 21.4 | 7.2 | 11.1 | 6.8 | 18.2 | 11.9 | 30.9 |
| | | 10.5 mV | 10.5 | 29.5 | 11.0 | 10.2 | 6.8 | 18.4 | 6.0 | 13.1 | 4.4 |
| DEMO | 64k | 8.0 pA | 7.4 | 2.6 | 21.3 | 7.2 | 8.0 | 7.0 | 51.1 | 11.9 | 9.8 |
| | | 13.4 mV | 13.4 | 16.1 | 29.2 | 11.3 | 8.6 | 13.5 | 8.2 | 11.2 | 13.6 |
| | 32k | 7.6 pA | 6.6 | 7.6 | 5.8 | 8.7 | 5.1 | 2.7 | 24.1 | 10.9 | 7.6 |
| | | 7.5 mV | 10.6 | 16.0 | 22.5 | 7.5 | 4.6 | 13.4 | 5.4 | 6.9 | 6.6 |
| NSGA2 | 64k | 4.9 pA | 4.9 | 3.2 | 3.7 | 1.2 | 9.5 | 3.1 | 24.7 | 13.7 | 6.9 |
| | | 2.7 mV | 3.2 | **2.5** | 3.0 | 2.7 | **3.0** | 1.8 | 4.5 | 2.6 | **2.1** |
| | 32k | 12.4 pA | 3.2 | 12.4 | 17.4 | 10.5 | 36.6 | 21.9 | 43.2 | 9.3 | 8.4 |
| | | **2.6 mV** | **3.2** | 2.9 | **2.5** | **2.6** | 3.4 | **1.8** | **4.4** | **2.6** | 2.2 |
| EP-GAN (ours) | 64k | 8.6 pA | 3.6 | 8.6 | 3.3 | 4.9 | 37.9 | 9.2 | 31.8 | 5.1 | 10.2 |

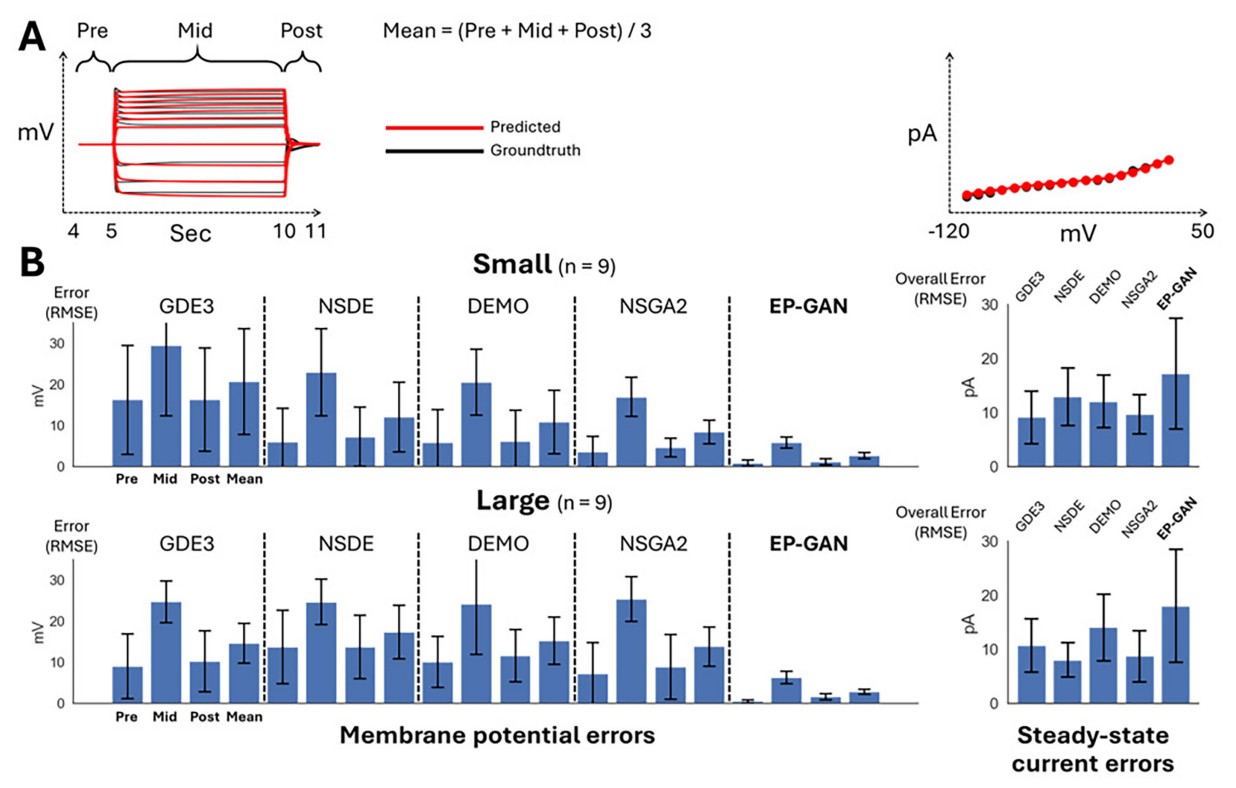

**Figure 5.** Bar plot showing the average root mean square error (RMSE) errors for membrane potential responses (pre-, mid-, post-activation periods, mean error) and steady-state currents for nine experimental neurons. (**A**) Membrane potential responses (left) and steady-state currents (right) diagrams showing the time and voltage intervals of which the RMSE errors are computed. (**B**) Bar plots showing RMSE errors (n = 9, 95% confidence level using t-test) for membrane potential responses and steady-state currents for small HH-model prediction scenarios (top) and large HH-model prediction scenarios (bottom). All methods use a 32k sample size for both scenarios.

## Parameter inference time

We also evaluate EP-GAN for its scalability by assessing its overall inference time and computational cost and comparing these to existing methods. Indeed, for estimation tasks involving many neurons, it is essential that the method is scalable so that the predictions are done within a reasonable time. In particular, for EP-GAN, the total time $T$ needed for modeling $N$ neurons including data generation and training time can be written as

$$T(N) \sim T_{Data\ generation} + T_{Train} + T_{Inference}$$

whereas for existing methods, the $T(N)$ follows the form

$$T(N) \sim N \cdot T_{Inference}$$

which increases linearly as $N$ increases. Since $T_{Inference}$ for EP-GAN is nearly instantaneous, it has a strong advantage in parameter prediction tasks involving multiple neurons. As an example, given a hypothetical task of modeling all 279 somatic neurons in the *C. elegans* nervous system, it would take DEMO, GDE3, NSDE, or NSGA2 more than 44 days (assuming 7.2k samples per neuron and our available computing environment) whereas, for EP-GAN, the process would be done within a day under a similar training setup. For a larger number of neurons, the computational requirement of existing methods would grow linearly while EP-GAN would require constant time to complete the inference. Such scalability largely benefits from EP-GAN learning a parameter estimation strategy that is applicable for multiple neuron classes and its neural network structure being an inherently parallel architecture, allowing it to take multiple neuron profiles and output corresponding parameters in a single forward pass (*Zou et al., 2009*).

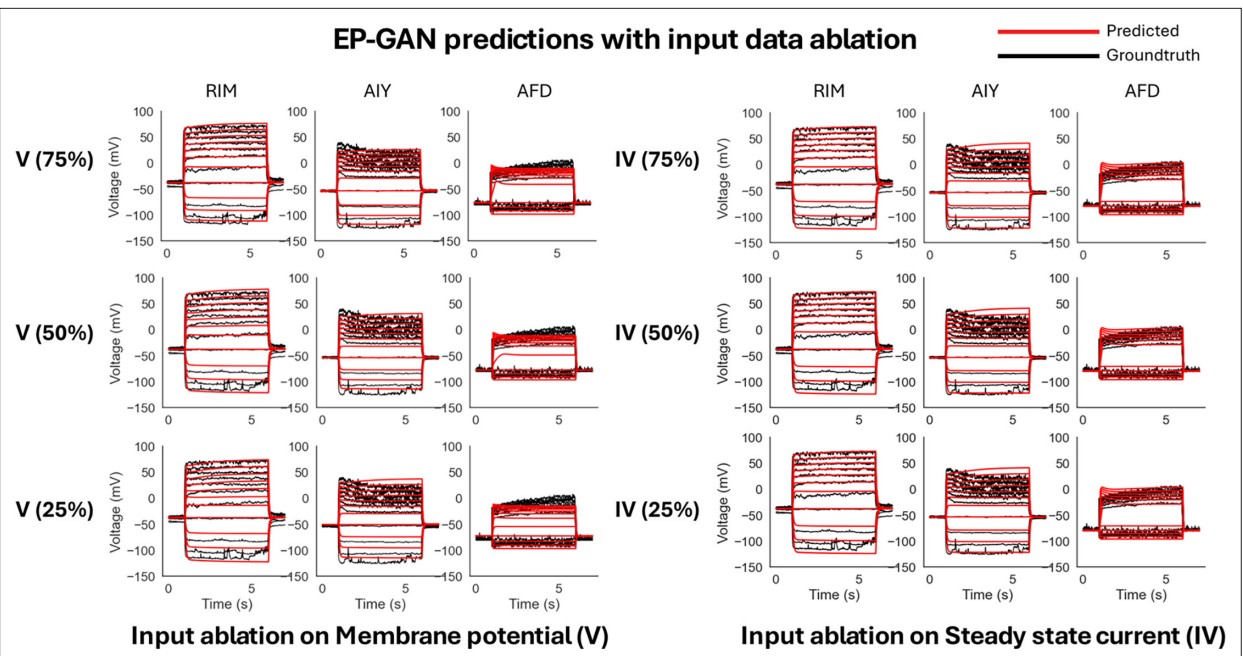

**Figure 6.** Input data ablation on ElectroPhysiomeGAN (EP-GAN) (32k, Large HH-model). Left: reconstructed membrane potential responses for RIM, AIY, and AFD when given with incomplete membrane potential responses data. Percentages in parentheses represent the remaining portion of input membrane potential responses trajectories. Right: reconstructed membrane potential responses for RIM, AIY, and AFD when given with incomplete steady-state current input.

The online version of this article includes the following figure supplement(s) for figure 6:

**Figure supplement 1.** Input data ablation on ElectroPhysiomeGAN (EP-GAN) (32k, Large HH-model).

## Discussion

In this work, we introduce a novel deep generative method and system called EP-GAN, for estimating HH-model parameters given the recordings of neurons with graded membrane potential (non-spiking). The proposed system encompasses the RNN encoder layer to process the neural recordings information such as membrane potential responses and steady-state current profiles and the Generator layer to generate a large number of HH-model parameters ($N > 175$). The system can be trained entirely on simulation data informed by an arbitrary HH-model. When applied to neurons in *C. elegans*, EP-GAN generates parameters of HH-model which membrane potential responses are closer to ground truth responses than the existing methods such as differential evolution and genetic algorithms. The advantage of EP-GAN is in the accuracy and inference speed achieved through fewer training samples than existing methods and is generic such that it does not depend on the number of neurons for which inference is to be performed (*Nicoletti et al., 2019*; *Naudin et al., 2022b*; *Nicoletti et al., 2024*). In addition, the method largely preserves performance when provided with input data with partial information such as missing membrane potential responses (up to 50% missing) or steady-state current traces (up to 75% missing).

While EP-GAN is a step forward toward the ElectroPhysiome model of *C. elegans*, its inability to support neurons with spiking membrane potential responses remains a limitation. The reason stems from the fact that neurons with spiking membrane potential responses are rare during the generation of training data of 16 ionic channels HH-model without the spike-associated neuron channels. The relative sparsity of spiking responses makes their translation strategies to parameter space difficult to learn. A similar limitation is present with bistable membrane potential responses, for example, AFD, AWB, and AWC, although to a lesser extent. While the limitations for these profiles can be partially remedied through more training samples of their neuron types, their relative sparseness in the training data tends to cause lower quality of predicted parameters. Indeed, previous studies of *C. elegans* nervous system found that the majority of neurons exhibit graded membrane potential response instead of spiking (*Nicoletti et al., 2019*; *Goodman et al., 1998*). Furthermore, the limitation could

**Table 3.** Ablation studies.

Top: membrane potential responses and steady-state current errors achieved for EP-GAN (32k, Large HH-model) when provided with incomplete input data. Bottom: membrane potential responses and steady-state current errors achieved for EP-GAN (32k, Large HH-model) upon using only adversarial loss (Adv) and using adversarial + current reconstruction loss (Adv + steady state) and all three loss components (Adv + steady state + membrane potential).

| Input Data Ablation | Sample size | Median Error | RIM | AIY | AFD |
|---|---|---|---|---|---|
| EP-GAN (25% membrane potential) | 32k | 5.4 mV<br>14.9 pA | 3.8<br>2.8 | 5.4<br>14.9 | 8.9<br>34.4 |
| EP-GAN (75% steady-state) | 32k | 3.5 mV<br>15.6 pA | 3.3<br>3.7 | 3.5<br>15.6 | 5.1<br>68.9 |
| EP-GAN (50% steady-state) | 32k | 3.5 mV<br>15.5 pA | 3.3<br>3.7 | 3.5<br>15.5 | 5.1<br>68.9 |
| EP-GAN (25% steady-state) | 32k | 3.5 mV<br>15.6 pA | 3.3<br>3.7 | 3.5<br>15.6 | 5.1<br>68.9 |
| EP-GAN (75% membrane potential) | 32k | 3.4 mV<br>11.3 pA | 3.4<br>3.6 | 2.7<br>11.3 | 4.9<br>44.0 |
| EP-GAN (full) | 32k | 3.3 mV<br>10.5 pA | 3.3<br>3.2 | 2.7<br>10.5 | 5.2<br>39.5 |
| EP-GAN (50% membrane potential) | 32k | 3.3 mV<br>13.5 pA | 3.3<br>3.4 | 2.9<br>13.5 | 4.5<br>43.2 |
| **Loss ablation** | **Sample size** | **Median Error** | **RIM** | **AIY** | **AFD** |
| EP-GAN (Adv) | 32k | 14.4 mV<br>20.3 pA | 14.4<br>3.1 | 6.1<br>20.3 | 24.5<br>75.4 |
| EP-GAN (Adv + steady state) | 32k | 6.0 mV<br>19.1 pA | 5.7<br>2.8 | 6.0<br>19.1 | 3.9<br>23.1 |
| EP-GAN (Adv + steady state + membrane potential) | 32k | 3.3 mV<br>10.5 pA | 3.3<br>3.2 | 2.7<br>10.5 | 5.2<br>39.5 |

lie within the current architecture of EP-GAN as it processes data directly without a component that discerns and processes spiking membrane potential responses. Improving the sampling strategy for training data alongside the enhancement of the network architecture could address these limitations in the future.

As discussed in 'Materials and methods', it is worth noting that EP-GAN does not necessarily recover the ground truth parameters that are associated with the input membrane potential responses and steady-state current profiles. This is mainly due to the fact that there may exist multiple parameter

**Table 4.** Simulation protocols for simulated and experimental neurons.

| Neuron | Duration (s) | Current-clamp (min:step:max) | Stimulation period (s) | Voltage-clamp (min:step:max) |
|---|---|---|---|---|
| Simulated | 15 | –15 pA:5 pA:35 pA | 5–10 | –120 mV:10 mV:50 mV |
| RIM | 15 | –15 pA:5 pA:35 pA | 5–10 | –120 mV:10 mV:50 mV |
| DVC | 15 | –2 pA:1 pA:8 pA | 5–10 | –120 mV:10 mV:50 mV |
| HSN | 15 | –2 pA:1 pA:8 pA | 5–10 | –120 mV:10 mV:50 mV |
| AIY | 15 | –15 pA:5 pA:35 pA | 5–10 | –120 mV:10 mV:50 mV |
| URX | 15 | –4 pA:2 pA:16 pA | 5–10 | –120 mV:10 mV:50 mV |
| RIS | 15 | –4 pA:2 pA:16 pA | 5–10 | –120 mV:10 mV:50 mV |
| AFD | 15 | –15 pA:5 pA:35 pA | 5–10 | –120 mV:10 mV:50 mV |
| AWB | 15 | –4 pA:2 pA:16 pA | 5–10 | –120 mV:10 mV:50 mV |
| AWC | 15 | –4 pA:2 pA:16 pA | 5–10 | –120 mV:10 mV:50 mV |

regimes for the HH-model that support the given inputs (*Willms, 2002*; *Marder and Goaillard, 2006*; *Marder and Taylor, 2011*; *Prinz et al., 2003*; *Prinz et al., 2004*; *Valle and Madureira, 2022*; *Raba et al., 2013*; *Naudin, 2023*). The parameters generated by a single forward pass of EP-GAN (i.e., a single flow of information from the input to the output) could thus be interpreted as a one-time sampling from such a regime, and a perturbation to inputs may result in a different set of parameters. Such sensitivity to perturbation could be adjusted by supplementing the training samples or inputs with additional recording data (e.g., multiple recording data per neuron).

EP-GAN allows additional modifications to accommodate different configurations of the problem. For instance, an update to the HH-model would only require retraining of the network without changes to its architecture. Indeed, the neuronal genome of *C. elegans* indicates additional voltage-gated channels that could be further incorporated into the HH-models introduced in *Nicoletti et al., 2019*; *Nicoletti et al., 2024* to improve its modeling accuracy of membrane potential dynamics (*Hobert, 2018*). Extending the inputs to include additional data, for example, channel activation profiles, can also be done in a straightforward manner by concatenating them to the input vectors of the Encoder network. Extending EP-GAN prediction capabilities to new neuron types can also be done by incorporating additional constraints during training data generation.

Despite its primary focus on *C. elegans* neurons, we believe EP-GAN and its future extensions could be viable for modeling a variety of neurons in other organisms. Indeed, there are increasing advances in resolving connectomes of more complex organisms and techniques for recording large-scale neural activities (*Brooks et al., 2022*; *Winding et al., 2023*; *Oh et al., 2014*; *Sofroniew et al., 2016*). As neurons in these organisms can be described by a generic HH-model or similar differential equation model, EP-GAN is expected to be applicable and contribute to the development of biologically detailed nervous system models of neurons in these organisms.

## Materials and methods

We divide the materials and methods section into three parts. In the first part, we describe the detailed architecture of the EP-GAN including its sub-modules with the simulation protocol used

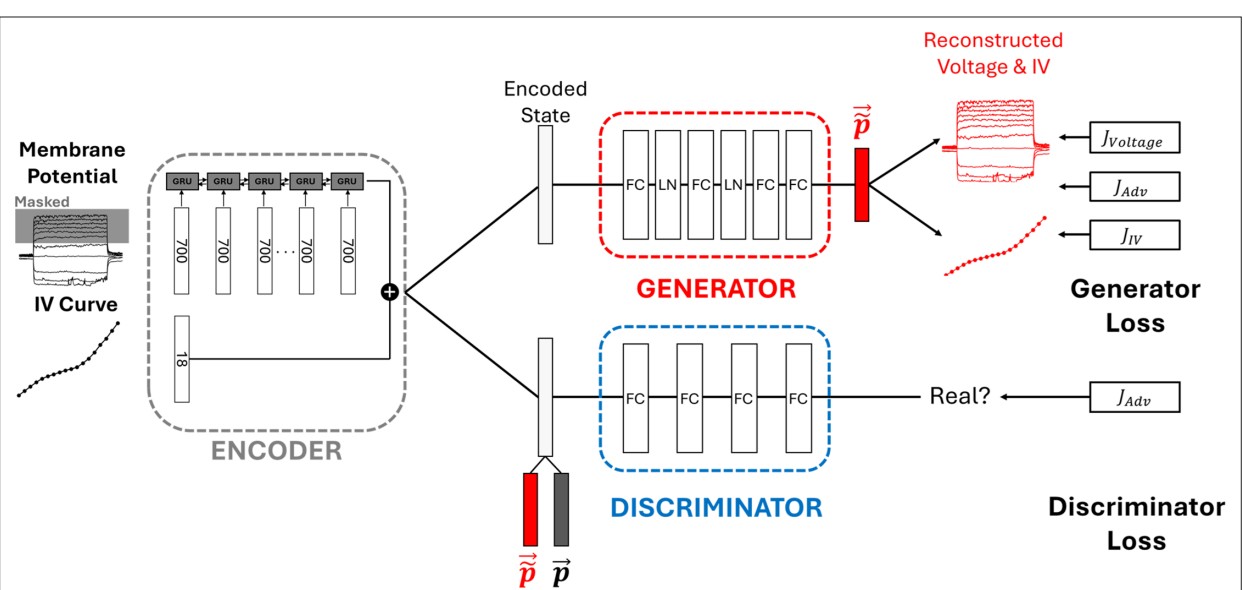

**Figure 7.** Architecture of ElectroPhysiomeGAN (EP-GAN). The architecture consists of an Encoder, Generator, and Discriminator. Encoder compresses the membrane potential responses into a 1D vector (i.e., latent space) that is then concatenated with 1D steady-state current profile to be used as an input to both Generator and Discriminator. Generator translates the latent space vector into a vector of parameters $\vec{\tilde{p}}$ and the Discriminator outputs a scalar measuring the similarity between generated parameters ($\vec{\tilde{p}}$) and ground truths ($\vec{p}$). The Generator is trained with adversarial loss supplemented by reconstruction losses for both membrane potential responses and steady-state current profiles. The Discriminator is trained with Discriminator adversarial loss only. Generator and Discriminator follow the architecture of Wasserstein GAN with gradient penalty (WGAN-GP) for more stable learning. During training, random masking is applied to input membrane potential responses where its masking rate gradually decreases as the training continues.

during training. In the second part, we describe the mathematical description for the ionic currents in the HH-model used for neuron modeling. In the third part, we describe the dataset and experimental protocol of novel neuron recordings of AWB, AWC, URX, RIS, DVC, and HSN from *C. elegans* nervous system.

## Architecture of EP-GAN

### Deep generative model for parameter prediction

EP-GAN receives neuronal recording data such as membrane potential responses and steady-state current profiles and generates a set of parameters that are associated with them in terms of simulating the inferred HH-model and comparing the simulated result with the inputs (*Figures 1 and 7*). We choose a deep generative model approach, specifically Generative Adversarial Network (GAN) as a base architecture of EP-GAN. The key advantage of GAN is in its ability to generate artificial data that closely resembles real data. The generative nature of GAN is advantageous for addressing the one-to-many nature of our problem, where there exist multiple parameter solutions for a given neuron recording. Indeed, several computational works attempting to solve an inverse HH-model noted the ill-posed nature of the parameter solutions (*Willms, 2002*; *Marder and Goaillard, 2006*; *Marder and Taylor, 2011*; *Prinz et al., 2003*; *Prinz et al., 2004*; *Valle and Madureira, 2022*; *Raba et al., 2013*; *Naudin, 2023*). Our approach is therefore leveraging GAN to learn a *domain of parameter sets* compatible with neuron recordings instead of mapping directly onto a single solution. GAN consists of two separate networks, Generator and Discriminator. The goal of the Generator is to generate outputs that are indistinguishable from real data, whereas the Discriminator's goal is to distinguish outputs that are generated by the Generator against real data. Throughout training, the Generator and the Discriminator engage in a zero-sum game until they both converge to optimal states (i.e. Nash equilibrium) (*Goodfellow et al., 2020*). The particular architecture we use is Wasserstein GAN with gradient penalty (WGAN-GP), a variant of GAN architecture offering more stable training and faster convergence (*Arjovsky et al., 2017*).

### Encoder module

In addition to Generator and Discriminator, we implement an Encoder module that pre-processes the input membrane potential responses for Generator and Discriminator (*Figure 7*, left). Specifically, the encoder serves two roles: (i) compression of membrane potential responses traces along the stimulus space, thus reducing its dimension from two-dimensional to one-dimensional, and (ii) translation of membrane potential responses traces into a latent space which encodes a meaningful internal representation for the Discriminator and Generator. The Encoder module uses Gated Recurrent Unit (GRU) architecture, a variant of RNN to perform this task (*Cho et al., 2014*). Each input sequence to a GRU cell at step $k$ corresponds to the entire membrane potential response of size 350 (i.e., 350 time points, representing $t = [4s, 11s]$) concatenated with the associated stimulus trace of equal size of 350. Since GRU is agnostic to the number of steps in an input sequence, such input structure allows EP-GAN to process a set of membrane potential traces with arbitrary current-clamp protocol. The output of the Encoder is a latent space vector of size 1024 encoding membrane potential responses information. The latent space vector is then concatenated with a 1D vector representing steady-state current profile which is used as an input to both Generator and Discriminator. During training, we randomly mask membrane potential traces to assist in better generalization in prediction (*Chang et al., 2022*; *Liu et al., 2024*). Specifically, we initially set the masking rate to 75% (i.e., 75% of membrane potential traces are randomly masked) and linearly decrease to 0% toward the end of training (*Figure 7*).

### Discriminator module

The goal of the Discriminator is given the input membrane potential responses and current profiles, to distinguish generated parameters from real ground truth parameters. The Discriminator receives as input the latent space vector from the Encoder concatenated with a generated or ground truth parameter vector and outputs a scalar representing the relative distance between two parameter sets (*Figure 7*, *Equation 1*). Such a quantity is called Wasserstein distance or Wasserstein loss and differs from a vanilla GAN Discriminator, which only outputs between 0 and 1. Wasserstein loss is known to remedy several common issues that arise from a vanilla GAN such as vanishing gradient and mode collapse, leading to more stable training (*Arjovsky et al., 2017*). To further improve the training of the

WGAN architecture, we supplement Wasserstein loss with a gradient penalty term, which ensures that the gradients of the Discriminator's output with respect to the input have unit norms (**Gulrajani et al., 2017**). This condition is called Lipschitz continuity and prevents Discriminator outputs from having large variations when there are only small variations in the inputs (**Gulrajani et al., 2017**; **Virmaux and Scaman, 2018**). Combined together, the Discriminator is trained with the following loss:

$$J_D = \mathbb{E}[D(\vec{\tilde{p}})] - \mathbb{E}[D(\vec{p})] + \lambda\mathbb{E}[(\|\nabla_{\hat{p}}D(\hat{p})\|_2 - 1)^2] \tag{1}$$

where $\mathbb{E}[D(\vec{\tilde{p}})]$ and $\mathbb{E}[D(\vec{p})]$ are the mean values of Discriminator outputs with respect to generated samples $\vec{\tilde{p}}$ and real samples $\vec{p}$, respectively, and $\lambda\mathbb{E}[(\|\nabla_{\hat{p}}D(\hat{p})\|_2 - 1)^2]$ is the gradient penalty term modulated by $\lambda$ where $\hat{p} = t\vec{\tilde{p}} + (1 - t)\vec{p}$ is the interpolation between generated and real samples with $0 \le t \le 1$.

## Generator module

Being an adversary network of Discriminator, the goal of the Generator is to fool the Discriminator by generating parameters that are indistinguishable from the real parameters. The Generator receives as input the concatenated vector from the Encoder and outputs a parameter vector (**Figure 7**). The module consists of four fully connected layers with layer normalization applied after the first two layers for improved model convergence (**Ba et al., 2016**). Each parameter in the output vector is scaled between –1 and 1, which is then scaled back to the parameters' original scales. The module is trained using three loss terms: (i) Generator adversarial loss, (ii) membrane potential responses reconstruction loss, and (iii) steady-state current reconstruction loss as follows:

$$J_G = -\mathbb{E}[D(\vec{\tilde{p}})] + J_{\vec{V}} + J_{\vec{IV}} \tag{2}$$

$$J_{\vec{V}} = \sum_{i=1}^{n} |\vec{V}_{groundtruth} - \vec{V}_{reconstructed}| \tag{3}$$

$$J_{\vec{IV}} = \sum_{i=1}^{n} |\vec{IV}_{groundtruth} - \vec{IV}_{reconstructed}| \tag{4}$$

where $-\mathbb{E}[D(\vec{\tilde{p}})]$ is Generator adversarial loss, which is the reciprocal of the mean Discriminator outputs with respect to generated samples, and $J_{\vec{V}}$ and $J_{\vec{IV}}$ are $L_1$ regression loss for reconstructed membrane potential responses and steady-state current profiles, respectively. It is important to note that $J_{\vec{V}}$ and $J_{\vec{IV}}$ are part of Generator's computation graph and thus force Generator to optimize them on top of adversarial loss (**Figure 8**). The composite loss function of Generator makes EP-GAN a 'model-informed' GAN as the HH-model itself becomes part of the training process. Such networks have been shown to be more data-efficient during training as they do not rely solely on training data to learn an effective strategy (**Karniadakis et al., 2021**; **Raissi et al., 2019**). The mathematical description of membrane potential responses and steady-state current reconstruction from generated parameter set $\vec{\tilde{p}}$ is as follows:

$$\vec{V}_{reconstructed}(t) = \nabla^{-1}\left(\frac{d\vec{V}}{dt}(\vec{V}, t, \vec{\tilde{p}})\right), \quad \vec{IV}_{reconstructed}(V) = IV(\vec{V}, \vec{\tilde{p}}) \tag{5}$$

Here, $\frac{d\vec{V}}{dt}(\vec{V}, t, \vec{\tilde{p}})$ is the right-hand-side function of the HH-model, which computes the membrane potential responses derivative at time $t$ given the membrane potential responses $\vec{V}$ and parameter set $\vec{\tilde{p}}$, and $IV(\vec{V}, \vec{\tilde{p}})$ is the function that evaluates the neuron's steady-state current $I$ given the voltage states $\vec{V}$ and parameter set $\vec{\tilde{p}}$. Membrane potential responses are reconstructed by first evaluating their derivatives with respect to ground truth membrane potential responses and generated parameters $\vec{\tilde{p}}$ at regularly sampled time points. This is followed by the forward integration operation $\nabla^{-1}$ similar to Euler's method to approximate $\vec{V}$ at sampled time points given the initial condition $\vec{V}_{init}$:

$$\vec{V}_{t+1} = \vec{V}_t + h\vec{V}'(t), \quad \vec{V}_{t=0} = \vec{V}_{init} \tag{6}$$

where $h$ is the time interval between sampled derivatives. $\vec{V}_{init}$ can be selected at any time point within the ground truth membrane potential state (e.g., pre-activation, mid-activation, post-activation) to reconstruct different membrane potential features. Notably, all computation steps consisting of the forward integration process are expected to be differentiable. This is necessary to incorporate the forward integration process as part of the generator network that requires full differentiability and thus is trainable via the back-propagation algorithm (*Rumelhart et al., 1986*). Computationally, we achieve this by manually implementing the forward integration process with discrete array operations that support auto-differentiation and vectorization (e.g., PyTorch Tensors) instead of simulating the membrane potential with ODE solvers (*Paszke et al., 2019*). The variable $h$ can also be adjusted to increase the accuracy of the membrane potential responses reconstruction in exchange for the increased computational cost. The current profile is reconstructed by directly evaluating $IV(\vec{V},\vec{p})$, which uses the generated parameter set $\tilde{\vec{p}}$ over the range of voltage values. We show in *Table 3* that reconstruction losses are essential for the accuracy of predicted parameters.

## Generating training data

For a successful training of a neural network model, the training data must be of a sufficient number of samples, denoised, and diverse. To ensure these conditions are met with a simulated dataset, we employ a three-step process for generating training data (*Figure 9*).

## Step 1

Randomly generate parameter sets by sampling each parameter within a predefined distribution. This distribution is the skewed normal distribution for channel conductance parameters and uniform distributions for other parameters. The range is determined according to the biologically feasible ranges reported in the literature (see table *predicted parameters* in *Supplementary file 1* for an explicit range used for each parameter) (*Liu et al., 2018*; *Naudin et al., 2021*; *Izhikevich, 2007*). In particular, search ranges for channel parameters are set to baseline ±50% where baseline parameters are default parameters given by *Nicoletti et al., 2024*.

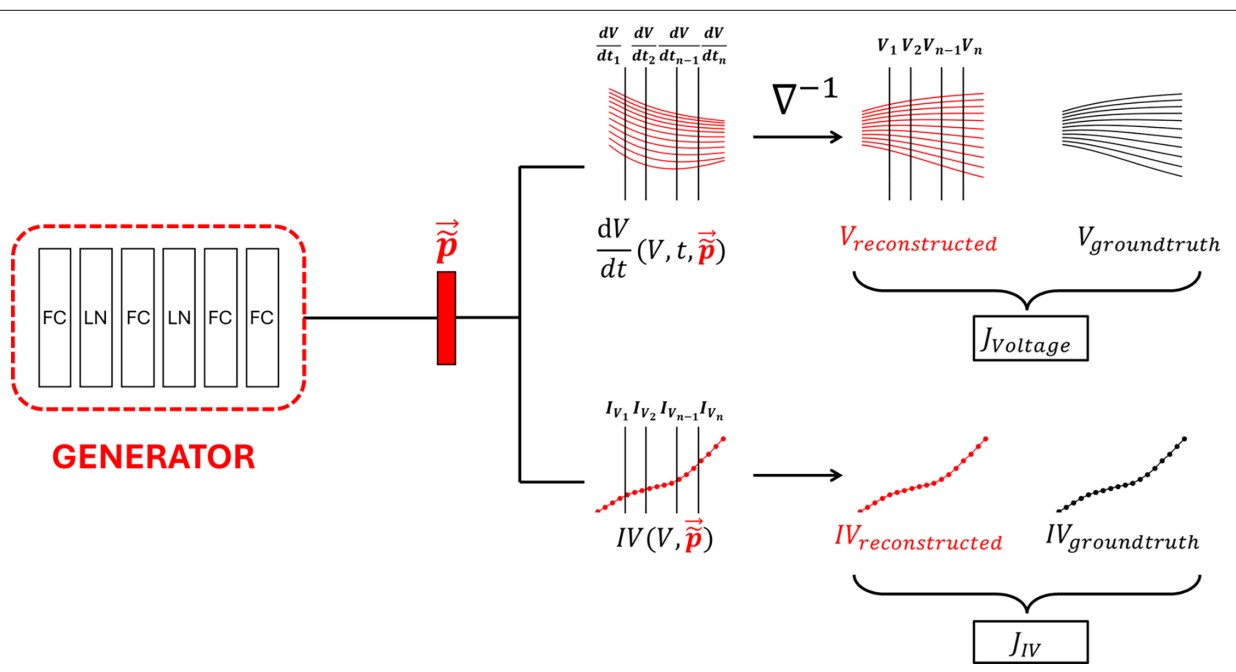

**Figure 8.** Description of membrane potential responses and steady-state current reconstruction losses for the Generator. Generated parameter vector $\tilde{\vec{p}}$ is used to evaluate membrane potential responses derivatives $dV/dt$ at $n$ time points sampled with fixed interval given the ground truth $\vec{V}$ at those time points. The evaluated membrane potential responses derivatives are then used to reconstruct $\vec{V}$ using the forward integration operation $\nabla^{-1}$. The reconstructed $\vec{V}$ is then compared with ground truth $\vec{V}$ to evaluate the membrane potential responses reconstruction loss $J_{\vec{V}}$. Steady-state current reconstruction is computed in a similar way via evaluating the currents at defined voltage points $V$ given generated parameters $\vec{p}$ as inputs.

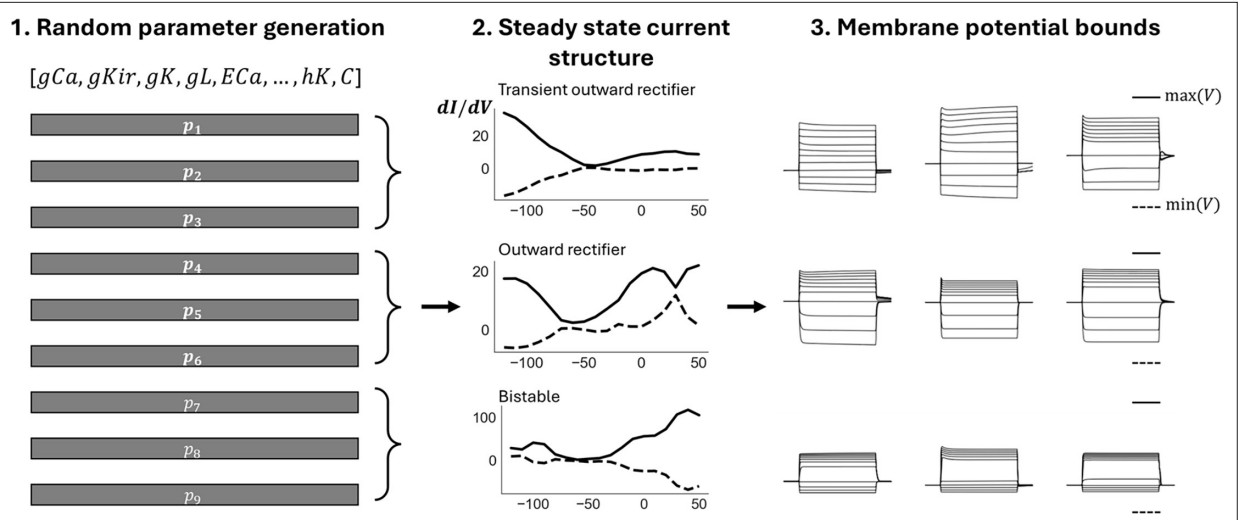

**Figure 9.** Training data generation. In Step 1, each parameter is initially sampled from biologically plausible ranges using both skewed Gaussian (channel conductance) and uniform sampling. A parameter set consists of 175 parameters spanning 16 known ion channels in *C. elegans* and similar organisms. In Steps 2 and 3, steady-state currents and membrane potential responses are evaluated for each parameter set to ensure they satisfy the predefined constraints such as bifurcation structure represented by *dI/dV* bounds and minimum-maximum membrane potential across current-clamp protocols. Only parameter sets that meet both constraints are included in the training set.

### Step 2

Simulate steady-state current traces for each sampled parameter set followed by imposing bifurcation structure constraints on each of them. This is done by calculating the first derivative of the currents *dI/dV* with respect to voltage states and ensuring they are within the 98% confidence intervals of experimentally obtained *dI/dV* bounds. Specifically, we evaluate each parameter set with respect to the three neuron types that are found within *C. elegans* non-spiking neurons – Type 1: transient outward rectifier (RIM, DVC, HSN); type 2: outward rectifier type (AIY, URX, RIS); and type 3: bistable type (AFD, AWB, AWC). During data generation, approximately the same number of neurons are generated for each type to ensure balance between all neuron types. The step can be further extended with new neuron types by incorporating additional *dI/dV* bounds.

### Step 3

Impose (minimum, maximum) constraints on the membrane potential response across the current-clamp protocol. These values are set to (–100 mV, 150 mV) respectively for (–15 pA, 35 pA) current-clamp range. Parameter sets that do not satisfy the steady-state currents (Step 2) and membrane potential responses constraints are then removed from the training set. These constraints serve two purposes: (i) remove parameter sets that result in non-realistic membrane potential responses/steady-state current profiles from the training set and (ii) serve as an initial data augmentation process for EP-GAN training. The constraints can also be extended or adjusted if deemed necessary for the improvement of the training process. Once constraints are applied, Gaussian noise is added to the membrane potential responses training data to mimic the measurement noises in experimental membrane potential responses recording data.

### Ionic currents modeling

The general equation describing the membrane potential dynamics of a single-compartment neuron is

$$C_m \frac{dV}{dt} = -I_{ion} + I_{Ext} \qquad (7)$$

where $I_{ion}$ and $I_{Ext}$ represent the ionic and external currents applied to a neuron, respectively. The HH-model we consider has a total of 16 ionic current terms comprised of 11 voltage/calcium-gated potassium currents ($I_{K^+}$), 3 voltage-gated calcium currents ($I_{Ca^{2+}}$), leakage currents ($I_{Leak}$), and sodium

**Table 5.** List of ion channels included in the estimated HH-models.
Ion selectivity for each channel and the number of ElectroPhysiomeGAN (EP-GAN) trained parameters (not including reversal potentials) for both small and large HH-models are listed.

| Ion channel | Ion selectivity | # of parameters (small HH-model) | # of parameters (large HH-model) |
|---|---|---|---|
| SHL1 | K⁺ | 4 | 22 |
| SHK1 | K⁺ | 3 | 14 |
| EGL2 | K⁺ | 2 | 8 |
| IRK1/3 | K⁺ | 2 | 10 |
| UNC103 | K⁺ | 3 | 15 |
| KQT1 | K⁺ | 3 | 17 |
| EXP2 | K⁺ | 3 | 15 |
| SLO1 | K⁺ | 2 | 2 |
| SLO1-CaV | K⁺ | 3 | 3 |
| SLO2 | K⁺ | 2 | 2 |
| SLO2-CaV | K⁺ | 3 | 3 |
| EGL19 | Ca⁺ | 3 | 23 |
| UNC2 | Ca⁺ | 3 | 18 |
| CCA1 | Ca⁺ | 3 | 15 |
| Leak | Leak | 1 | 1 |
| NCA | Na⁺ | 1 | 1 |

leakage currents ($I_{Na^+}$) (**Table 5**). Consequently, the ionic current term $I_{ion}$ of the considered HH-model can be written as

$$I_{ion} = I_{K^+} + I_{Ca^{2+}} + I_{Leak} + I_{Na^+} \qquad (8)$$

Each $ith$ ionic current can be modeled according to the HH-type formulation as follows:

$$I^i_{ion} = g_i \cdot m_i^p \cdot h_i^p \cdot (V - E_{rev}) \qquad (9)$$

where $g_i$ is the maximum conductance and $E_{rev}$ is the reversal potential of the channel. $m_i^p$ and $h_i^p$ each represent the activation and inactivation variables of the channel with the following equations:

$$\frac{dm_i}{dt} = \frac{m_{i,\infty} - m_i}{\tau_{i,m}} \qquad (10)$$

$$\frac{dh_i}{dt} = \frac{h_{i,\infty} - h_i}{\tau_{i,h}} \qquad (11)$$

where $m_{i,\infty}$ and $h_{i,\infty}$ each represent the steady-state activation and inactivation values, and $\tau_{i,m}$ and $\tau_{i,h}$ represent the activation and inactivation time constants. Note that each ionic channel may have a different mathematical description and parameters for variables $m_{i,\infty}$, $h_{i,\infty}$, $\tau_{i,m}$, and $\tau_{i,h}$ according to their dynamics and dependencies. While the majority of these variables are dependent on membrane potential, some channels are dependent on intracellular calcium (e.g., SLO1/2) or independent (e.g., $\tau_{i,h}$ of SHK1).

Leakage and sodium leakage currents are modeled with the following equations:

$$I_{Leak} = g_{Leak}(V - E_{Leak}) \qquad (12)$$

$$I_{Na^+} = g_{NCA}(V - E_{NCA}) \qquad (13)$$

where $g_{Leak}$ and $g_{NCA}$ each represent the conductance values for the leakage and sodium leakage currents. For the exact mathematical equations describing each individual channel, please refer to the supplementary materials of *Nicoletti et al., 2024*.

## Experimental protocol

### *C. elegans* culture and strains

All animals used in this study were maintained at room temperature (22°C–23°C) on nematode growth medium (NGM) plates seeded with *Escherichia coli* OP50 bacteria as a food source (*Brenner, 1974*). The strains used in this study were CX7893 kyIs405 (AWB), CX3695 kyIs140 (AWC), ZG611 iaIs19 (URX), EG1285 lin-15B(n765);oxIs12 (RIS), UL2650 leEx2650 (DVC), and CX4857 kyIs179 (HSN).

### Electrophysiology

Electrophysiological recordings were performed on young adult hermaphrodites (~3 days old) at room temperature as previously described (*Liu et al., 2018*). The gluing and dissection were performed under an Olympus SZX16 stereomicroscope equipped with a 1× Plan Apochromat objective and widefield 10× eyepieces. Briefly, an adult animal was immobilized on a Sylgard-coated (Sylgard 184, Dow Corning) glass coverslip in a small drop of DPBS (D8537; Sigma) by applying a cyanoacrylate adhesive (Vetbond tissue adhesive; 3M) along one side of the body. A puncture in the cuticle away from the incision site was made to relieve hydrostatic pressure. A small longitudinal incision was then made using a diamond dissecting blade (type M-DL 72029 L; EMS) along the glue line adjacent to the neuron of interest. The cuticle flap was folded back and glued to the coverslip with GLUture Topical Adhesive (Abbott Laboratories), exposing the neuron to be recorded. The coverslip with the dissected preparation was then placed into a custom-made open recording chamber (~1.5 ml volume) and treated with 1 mg/ml collagenase (type IV; Sigma) for ~10 s by hand pipetting. The recording chamber was subsequently perfused with the standard extracellular solution using a custom-made gravity-feed perfusion system for ~10 ml.

All electrophysiological recordings were performed with the bath at room temperature under an upright microscope (Axio Examiner; Carl Zeiss, Inc) equipped with a 40× water immersion lens and 16× eyepieces. Neurons of interest were identified by fluorescent markers and their anatomical positions. Preparations were then switched to the differential interference contrast (DIC) setting for patch-clamp. Electrodes with resistance (RE) of 15–25 MΩ were made from borosilicate glass pipettes (BF100-58-10; Sutter Instruments) using a laser pipette puller (P-2000; Sutter Instruments) and fire-polished with a microforge (MF-830; Narishige). We used a motorized micromanipulator (PatchStar Micromanipulator; Scientifica) to control the electrodes back filled with standard intracellular solution. The standard pipette solution was (all concentrations in mM) [K-gluconate 115; KCl 15; KOH 10; MgCl$_2$ 5; CaCl$_2$ 0.1; Na$_2$ATP 5; NaGTP 0.5; Na-cGMP 0.5; cAMP 0.5; BAPTA 1; HEPES 10; sucrose 50], with pH adjusted with KOH to 7.2, osmolarity 320–330 mOsm. The standard extracellular solution was [NaCl 140; NaOH 5; KCl 5; CaCl$_2$ 2; MgCl$_2$ 5; sucrose 15; HEPES 15; dextrose 25], with pH adjusted with NaOH to 7.3, osmolarity 330–340 mOsm. Liquid junction potentials were calculated and corrected before recording. Whole-cell current clamp and voltage-clamp experiments were conducted on an EPC-10 amplifier (EPC-10 USB; Heka) using PatchMaster software (Heka). Two-component capacitive compensation was optimized at rest, and series resistance was compensated to 50%. Analog data were filtered at 2 kHz and digitized at 10 kHz. Current-injection and voltage-clamp steps were applied through the recording electrode.

## Data and code availability

The membrane potentials and steady-state currents recording data for nine experimental neurons can be found in Mendeley Data, https://doi.org/10.17632/g5kcjp7jsk.1.

Generated HH-model parameters of all nine experimental neurons considered in the study (small and large HH-models, using EP-GAN (32k)) as well as minimum and maximum values for each parameter used during inference are deposited in supporting files 1.

Pre-trained PyTorch EP-GAN (64k) models for both small and large HH-models and the Jupyter Notebook testing the models with respect to nine experimental *C. elegans* neurons studied in the paper are available at the Github repository (https://github.com/shlizee/epgan copy archived at *Shlizerman, 2025*).

## Acknowledgements

This work was supported in part by National Science Foundation grant CRCNS IIS-2113003 (JK,ES), Washington Research Fund (ES), CRCNS IIS-2113120 (QL), Kavli NSI Pilot Grant (QL), CityU New Research Initiatives/Infrastructure Support from Central APRC 9610587 (QL), the General Research Fund (GRF) and Early Career Scheme (ECS) Award from Hong Kong Research Grants Council RGC (CityU 21103522, CityU 11104123, CityU 11100524) (QL), and Chan Zuckerberg Initiative (to Cori Bargmann). The authors also acknowledge the partial support by the Departments of Electrical and Computer Engineering (JK, ES), Applied Mathematics (ES), the Center of Computational Neuroscience (ES), and the eScience Institute (ES, JK) at the University of Washington. In addition, we thank Cori Bargmann and Ian Hope for *C. elegans* strains. Some strains were provided by the CGC, which is funded by the NIH Office of Research Infrastructure Programs (P40 OD010440). We thank Saba Heravi for discussions regarding parameter inference for electrophysiological recordings.

## Additional information

### Funding

| Funder | Grant reference number | Author |
|---|---|---|
| National Science Foundation | CRCNS IIS-2113003 | Jimin Kim<br>Eli Shlizerman |
| National Science Foundation | CRCNS IIS-2113120 | Qiang Liu |
| The Kavli Foundation | | Qiang Liu |
| City University of Hong Kong | | Qiang Liu |
| CityU New Research Initiatives | 9610587 | Qiang Liu |
| Hong Kong Research Grants Council RGC | CityU 21103522 | Qiang Liu |
| Hong Kong Research Grants Council RGC | CityU 11104123 | Qiang Liu |
| Hong Kong Research Grants Council RGC | CityU 11100524 | Qiang Liu |
| NIH Office of Research Infrastructure Programs | P40 OD010440 | Qiang Liu |

The funders had no role in study design, data collection and interpretation, or the decision to submit the work for publication.

### Author contributions

Jimin Kim, Conceptualization, Software, Formal analysis, Validation, Investigation, Visualization, Methodology, Writing - original draft, Project administration, Writing – review and editing; Minxian Peng, Shuqi Chen, Data curation; Qiang Liu, Data curation, Supervision, Funding acquisition, Writing – review and editing; Eli Shlizerman, Conceptualization, Resources, Supervision, Funding acquisition, Methodology, Project administration, Writing – review and editing

### Author ORCIDs

Jimin Kim ⬥ https://orcid.org/0000-0002-5597-5142
Minxian Peng ⬥ https://orcid.org/0000-0003-0419-3543
Shuqi Chen ⬥ https://orcid.org/0009-0003-9744-6964
Qiang Liu ⬥ https://orcid.org/0000-0002-9232-1420
Eli Shlizerman ⬥ https://orcid.org/0000-0002-3136-4531

Reviewer #2 (Public review): https://doi.org/10.7554/eLife.95607.4.sa1

Author response https://doi.org/10.7554/eLife.95607.4.sa2

## Additional files

### Supplementary files
MDAR checklist

Supplementary file 1. Table of EP-GAN(32k) generated parameters (Small, Large).

### Data availability
Membrane potential dynamics and steady-state current responses of all 9 experimental neurons considered in the study are deposited in Mendeley Data. Generated HH-model parameters of all 9 experimental neurons considered in the study (small and large HH-models), minimum and maximum values for each parameter used during inference, and parameter values predicted by EP-GAN are deposited in *Supplementary file 1*. Pre-trained EP-GAN models (64k sample size) for estimating small and large HH-models are deposited in Github.

The following dataset was generated:

| Author(s) | Year | Dataset title | Dataset URL | Database and Identifier |
|---|---|---|---|---|
| Peng M, Chen S, Liu Q | 2025 | ElectroPhysiomeGAN: Generation of Biophysical Neuron Model Parameters from Recorded Electrophysiological Responses | https://doi.org/10.17632/g5kcjp7jsk.1 | Mendeley Data, 10.17632/g5kcjp7jsk.1 |

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

# Appendix 1

## Error calculation

The overall errors between predicted membrane potential traces and their ground truth counterparts are computed using RMSE formula as follows:

$$V_{error} = \sqrt{\frac{\sum_{t=1}^{T}\left(\sum_{k=1}^{N_{stim}}(V_{k,t}^{pred} - V_{k,t}^{gt})^2/N_{stim}\right)}{N_T}}$$

Voltage with upper subscripts *pred* and *gt* represents reconstructed membrane potential values at trace $k$ and time $t$ using predicted HH-parameters and ground truth variables, respectively. $N_{stim}$ corresponds to a total number of stimulus values associated with current-clamp protocol. $N_T$ represents the total number of time points in which voltage traces are defined.

The error is computed for each of three intervals: pre-activation [4–5 s], mid-activation [5–10 s], and post-activation (10–11 s), which are then averaged to compute the overall error.

**Appendix 1—table 1.** Root mean square error (RMSE) errors for membrane potential responses (top) and steady-state currents (bottom) for test neurons (n=200) considered in prediction on simulated neurons scenario.

Membrane potential responses errors are ordered as pre-activation error [4–5 s], mid-activation error [5–10 s], and post-activation periods error (10–11 s).

| Method | Simulation # | Test neurons |
|---|---|---|
| EPGAN | 32k | 1.33 mV \| 3.63 mV \| 2.15 mV |
| | | 8.41 pA |

## Numerical simulation of HH-model

For efficient simulations of HH-model membrane potential dynamics, we use Julia ODE solver KenCarp47 algorithm supplemented by high-performance computing packages such as NumPy and SciPy (*Rackauckas and Nie, 2017*; *Harris et al., 2020*; *Virtanen et al., 2020*). Both relative and absolute tolerances for the ODE solver have been set to 1e-8 to ensure the accuracy of the simulations.

**Appendix 1—table 2.** Small HH-model root mean square error (RMSE) errors (sample size = 32k) for membrane potential responses and steady-state currents for *predictions on small HH-model scenarios.*

For each neuron, top row shows the RMSE errors for three time intervals – pre-activation [4–5s], mid-activation [5–10s], and post-activation (10–11s] and bottom row shows the RMSE error for steady-state currents across 18 voltage points.

| Method | Neurons | | | | | | | | |
|---|---|---|---|---|---|---|---|---|---|
| | RIM | | | DVC | | | HSN | | |
| | 5.63 mV | 53.09 mV | 3.92 mV | 47.39 mV | 79.42 mV | 47.34 mV | 2.8 mV | 14.76 mV | 8.98 mV |
| | 5.78 pA | | | 5.78 pA | | | 19.84 pA | | |
| | AIY | | | URX | | | RIS | | |
| | 9.75 mV | 17.41 mV | 11.75 mV | 14.89 mV | 25.19 mV | 10.61 mV | 28.39 mV | 19.52 mV | 28.18 mV |
| | 4.82 pA | | | 2.42 pA | | | 5.94 pA | | |
| | AFD | | | AWB | | | AWC | | |
| | 0.66 mV | 17.37 mV | 0.93 mV | 36.72 mV | 26.7 mV | 34.21 mV | 0.56 mV | 12.42 mV | 1.2 mV |
| GDE3 | 19.13 pA | | | 7.22 pA | | | 11.92 pA | | |

*Appendix 1—table 2 Continued on next page*

*Appendix 1—table 2 Continued*

| Method | Neurons | | | | | | | | |
|---|---|---|---|---|---|---|---|---|---|
| | RIM | | | DVC | | | HSN | | |
| | 33.29 mV | 51.33 mV | 31.54 mV | 0.86 mV | 19.88 mV | 4.29 mV | 9.76 mV | 41.66 mV | 9.22 mV |
| | 3.14 pA | | | 21.47 pA | | | 9.7 pA | | |
| | AIY | | | URX | | | RIS | | |
| | 0.46 mV | 11.84 mV | 4.69 mV | 1.35 mV | 15.19 mV | 4.61 mV | 5.86 mV | 21.26 mV | 5.96 mV |
| | 13.63 pA | | | 14.2 pA | | | 5.68 pA | | |
| | AFD | | | AWB | | | AWC | | |
| | 1.74 mV | 13.16 mV | 1.49 mV | 0.5 mV | 15.88 mV | 2.39 mV | 0.74 mV | 17.47 mV | |
| NSDE | 24.73 pA | | | 9.55 pA | | | 14.49 pA | | |
| | RIM | | | DVC | | | HSN | | |
| | 32.73 mV | 43.2 mV | 31.84 mV | 7.92 mV | 27.06 mV | 7.24 mV | 1.18 mV | 14.5 mV | 1.59 mV |
| | 6.52 pA | | | 13.77 pA | | | 14.63 pA | | |
| | AIY | | | URX | | | RIS | | |
| | 7.26 mV | 26.04 mV | 6.27 mV | 1.7 mV | 22.14 mV | 3.09 mV | 1.29 mV | 15.55 mV | 3.26 mV |
| | 5.16 pA | | | 5.39 pA | | | 9.68 pA | | |
| | AFD | | | AWB | | | AWC | | |
| | 0.85 mV | 13.06 mV | 1.08 mV | 0.05 mV | 14.41 mV | 0.21 mV | 0.8 mV | 9.44 mV | 1.23 mV |
| DEMO | 23.84 pA | | | 11.47 pA | | | 18.7 pA | | |
| | RIM | | | DVC | | | HSN | | |
| | 3.1 mV | 26.28 mV | 7.88 mV | 16.15 mV | 21.63 mV | 8.39 mV | 0.48 mV | 6.67 mV | 0.64 mV |
| | 3.97 pA | | | 7.39 pA | | | 14.29 pA | | |
| | AIY | | | URX | | | RIS | | |
| | 3.58 mV | 22.2 mV | 3.71 mV | 2.36 mV | 10.37 mV | 5.46 mV | 2.88 mV | 13.94 mV | 2.36 mV |
| | 4.59 pA | | | 16.55 | | | 5.01 | | |
| | AFD | | | AWB | | | AWC | | |
| | 2.24 mV | 17.99 mV | 2.37 mV | 0.52 mV | 19.49 mV | 8.32 mV | 1.9 mV | 14.82 mV | 2.98 mV |
| NSGA2 | 14.38 pA | | | 11.86 pA | | | 9.98 pA | | |
| | RIM | | | DVC | | | HSN | | |
| | 0.33 mV | 8.23 mV | 1.52 mV | 0.19 mV | 5.74 mV | 1.22 mV | 0.18 mV | 4.02 mV | 0.49 mV |
| | 4.03 pA | | | 13.8 pA | | | 10.29 pA | | |
| | AIY | | | URX | | | RIS | | |
| | 0.66 mV | 6.41 mV | 0.55 mV | 1.66 mV | 5.07 mV | 2.82 mV | 0.74 mV | 3.77 mV | 0.59 mV |
| | 10.7 pA | | | 16.84 pA | | | 13.8 pA | | |
| | AFD | | | AWB | | | AWC | | |
| | 3.15 mV | 8.66 mV | 2.87 mV | 0.04 mV | 7.23 mV | 0.33 mV | 0.66 mV | 4.71 mV | 0.72 mV |
| EP-GAN | 47.97 pA | | | 9.64 pA | | | 28.86 pA | | |

**Appendix 1—table 3.** Large HH-model root mean square error (RMSE) errors (sample size = 32k) for membrane potential responses and steady-state currents for *predictions on large HH-model scenarios*.

For each neuron, the top row shows the RMSE errors for three time intervals – pre-activation [4–5s], mid-activation [5–10s], and post-activation (10–11s], and the bottom row shows the RMSE error for steady-state currents across 18 voltage points.

| Method | Neurons | | | | | | | | | | |
|---|---|---|---|---|---|---|---|---|---|---|---|
| | RIM | | | | DVC | | | | HSN | | |
| | 2.31 mV | 30.01 mV | 9.71 mV | | 3.08 mV | 27.0 mV | 7.54 mV | | 3.4 mV | 31.0 mV | 4.09 mV |
| | 3.15 pA | | | | 23.53 pA | | | | 12.75 pA | | |
| | AIY | | | | URX | | | | RIS | | |
| | 16.88 mV | 25.85 mV | 15.51 mV | | 3.79 mV | 20.27 mV | 2.97 mV | | 6.54 mV | 32.36 mV | 6.61 mV |
| | 10.24 pA | | | | 6.32 pA | | | | 5.99 pA | | |
| | AFD | | | | AWB | | | | AWC | | |
| | 33.71 mV | 20.5 mV | 33.86 mV | | 3.8 mV | 24.56 mV | 4.14 mV | | 8.05 mV | 11.58 mV | 8.04 mV |
| GDE3 | 7.85 pA | | | | 18.08 pA | | | | 9.55 pA | | |
| | RIM | | | | DVC | | | | HSN | | |
| | 27.97 mV | 40.68 mV | 25.98 mV | | 13.33 mV | 30.91 mV | 12.46 mV | | 5.03 mV | 15.31 mV | 5.73 mV |
| | 7.75 pA | | | | 8.03 pA | | | | 9.58 pA | | |
| | AIY | | | | URX | | | | RIS | | |
| | 6.81 mV | 22.46 mV | 7.05 mV | | 0.19 mV | 21.14 mV | 4.45 mV | | 24.54 mV | 22.72 mV | 24.22 mV |
| | 6.82 pA | | | | 5.06 pA | | | | 4.17 pA | | |
| | AFD | | | | AWB | | | | AWC | | |
| | 24.96 mV | 2.09 mV | 18.19 mV | | 31.41 mV | 21.6 mV | 28.39 mV | | 12.25 mV | 22.77 mV | 13.38 mV |
| NSDE | 14.73 pA | | | | 7.24 pA | | | | 6.18 pA | | |
| | RIM | | | | DVC | | | | HSN | | |
| | 8.91 mV | 63.94 mV | 11.11 mV | | 16.01 mV | 21.32 mV | 15.96 mV | | 11.8 mV | 26.0 mV | 12.11 mV |
| | 7.57 pA | | | | 11.89 pA | | | | 21.4 pA | | |
| | AIY | | | | URX | | | | RIS | | |
| | 1.46 mV | 25.9 mV | 3.21 mV | | 23.15 mV | 18.16 mV | 27.24 mV | | 19.32 mV | 15.3 mV | 19.8 mV |
| | 7.24 pA | | | | 11.14 pA | | | | 6.81 pA | | |
| | AFD | | | | AWB | | | | AWC | | |
| | 2.29 mV | 11.92 mV | 3.79 mV | | 7.0 mV | 24.03 mV | 8.3 mV | | 1.03 mV | 10.97 mV | 3.22 mV |
| DEMO | 18.17 pA | | | | 11.9 pA | | | | 30.85 pA | | |
| | RIM | | | | DVC | | | | HSN | | |
| | 3.58 mV | 32.89 mV | 3.63 mV | | 7.55 mV | 33.6 mV | 7.06 mV | | 31.5 mV | 24.36 mV | 31.81 mV |
| | 6.57 pA | | | | 7.63 pA | | | | 5.75 pA | | |
| | AIY | | | | URX | | | | RIS | | |
| | 0.38 mV | 14.99 mV | 18.39 mV | | 3.61 mV | 20.4 mV | 1.87 mV | | 11.87 mV | 17.09 mV | 11.45 mV |
| | 8.68 pA | | | | 5.13 pA | | | | 2.74 pA | | |
| | AFD | | | | AWB | | | | AWC | | |
| | 0.8 mV | 22.68 mV | 1.14 mV | | 1.14 mV | 30.38 mV | 2.11 mV | | 5.12 mV | 32.47 mV | 3.17 mV |
| NSGA2 | 24.07 pA | | | | 10.9 pA | | | | 7.55 pA | | |

| Method | Neurons | | | | | | | | |
|---|---|---|---|---|---|---|---|---|---|
| | RIM | | | DVC | | | HSN | | |
| | 0.24 mV | 7.78 mV | 1.65 mV | 0.30 mV | 5.90 mV | 1.39 mV | 0.46 mV | 6.62 mV | 2.02 mV |
| | 3.21 pA | | | 12.35 pA | | | 17.44 pA | | |
| | AIY | | | URX | | | RIS | | |
| | 0.43 mV | 6.57 mV | 1.12 mV | 0.74 mV | 4.58 mV | 3.78 mV | 0.27 mV | 3.42 mV | 1.78 mV |
| | 10.52 pA | | | 36.64 pA | | | 21.86 pA | | |
| | AFD | | | AWB | | | AWC | | |
| | 1.68 mV | 9.99 mV | 1.86 mV | 0.37 mV | 7.24 mV | 0.31 mV | 0.57 mV | 4.93 mV | 0.84 mV |
| EP-GAN | 43.20 pA | | | 9.27 pA | | | 8.35 pA | | |

