## [Editor Report · eLife Assessment]

This study is a **valuable** contribution to the field of neuronal modeling by way of providing a method for rapidly obtaining neuronal physiology parameters from electrophysiological recordings. The method is **solid** as the generated models reproduce both ground-truth simulated data and empirical data, and there is now a quantitative comparison with other approaches.

---

## [Referee Report · Reviewer #2 (Public review)]

Summary:

Developing biophysically detailed computational models that accurately capture the characteristic physiological properties of neurons across diverse cell types is a key challenge in computational neuroscience. A major obstacle lies in determining the large number of model parameters, which are notoriously difficult to fit such that the model faithfully reproduces the empirically observed electrophysiological responses. Existing approaches require substantial computational resources to generate models for even a single neuron. Generating models for additional neurons typically requires starting from scratch, with no reuse of previous computations - making the process just as computationally expensive each time.

Kim et al. introduce an innovative approach based on a Generative Adversarial Network (GAN) to overcome these limitations. Once trained, the network takes empirically observed electrophysiological responses as input and predicts the biophysical parameters with which a Hodgkin-Huxley model can reproduce these responses. The authors demonstrate this for nine non-spiking neurons in *C. elegans*. The resulting models generally provide a good fit to the empirical data. As the GAN has learned general relationships between biophysical parameters and the resulting electrophysiology, it can be used to generate models of diverse cell types without retraining - enabling model generation at low computational cost.

Strengths:

The authors address an important and technically challenging problem. A noteworthy strength of their approach is that, once trained, the GAN can generate models from new empirical data at low computational cost. The generated models reproduce the responses to current injections well.

The authors have addressed all of my previous major concerns and have significantly improved their method:

(1) Most importantly, the generated models reproduce both ground-truth simulated and empirical data well. Responses - including resting membrane potentials - are now well captured.

(2) The comparison with other approaches has been extended to be more quantitative and rigorous.

(3) The authors now convincingly demonstrate that the improved EP-GAN is relatively robust to data ablation.

Weaknesses:

Slow dynamics (e.g., slow ramps) are still not reliably captured. However, as the approach excels at other frontiers - the generation of models for diverse cell types at low computational cost - I consider this to be a relatively minor limitation.

---

## [Author Response]

The following is the authors’ response to the previous reviews

**Reviewer #1 (Public review):**
(1) The bad equilibria of the model still remain a concern, as well as other features like the transient overshoots that do not match with the data. I think they could achieve more accuracy here by assigning more weight to such specific features, through adding these as separate objectives for the generator explicitly. The traces contain a five-second current steps, and one second before and one second after the training step. This means that in the RMSE, the current step amplitude will dominate as a feature, as this is simply the state for which the data trace contains most time-points. Note that this is further exacerbated by using the IV curve as an auxiliary objective. I believe a better exploration of specific response features, incorporated as independently weighted loss terms for the generator, could improve the fit. E.g. an auxiliary term could be the equilibrium before and after the current step, another term could penalise response traces that do not converge back to their initial equilibrium, etc.

We thank the reviewer for the suggestion. We supplemented the membrane potential regression loss with errors computed for 3 intervals: pre- post- and mid- stimulation time intervals, improving the accuracy of EP-GAN for baseline membrane potential responses (Figure 2, 3, Table S2, S3). We also changed the simulation protocols for generated parameters by allowing a longer simulation time of 15 seconds, where the stimulation is applied during [5, 10] seconds and no stimulation at t = [0, 5) (pre-stimulation) and t = (10, 15] (post-stimulation). These time intervals are chosen to ensure sufficient stabilization periods before and after stimulation.

(2) The explanation of what the authors mean with 'inverse gradient operation' is clear now. However, this term is mathematically imprecise, as the inverse gradient does not exist because the gradient operator is not injective. The method is simply forward integration under the assumption that the derivate of the voltage is known at the grid time-points, and should be described as such.

We thank the reviewer for the clarification on inverse gradient operation terminology. In the Methods section, we changed the term describing the inverse gradient operation to ‘forward integration’ which is a more accurate description describing the process.

(3) I appreciate that the authors' method provides parameters of models at a minimal computational cost compared to running an evolutionary optimization for every new recording. I also believe that with some tweaking of the objective, the method could improve in accuracy. However, I share reviewer 2's concerns that the evolutionary baseline methods are not sufficiently explored, as these methods have been used to successfully fit considerably more complex response patterns. One way out of the dilemma is to show that the EP-GAN estimated parameters provide an initial guess that considerably narrows the search space for the evolutionary algorithm. In this context, the authors should also discuss the recent gradient based methods such as Deistler et al. (https://doi.org/10.1101/2024.08.21.608979) or Jones et al (https://doi.org/10.48550/arXiv.2407.04025).

We supplemented the optimization setup for existing methods (GDE3, NSDE, DEMO, and NSGA2) by incorporating steady-state response constraints as the initial selection process. The process is similar to that of EP-GAN training data generation and DEMO parameter selection process [16] (see Results section, page 6 for detail). We also expanded the testing scenarios by evaluating all methods with respect to both small and large HH-model estimation. The small HH-model scenario estimates 47 parameters consisting of channel conductance, reversal potentials and initial conditions with the channel parameters (n = 129) frozen to default values in [41]. Large HH-model includes estimating channel parameters (i.e. 129) in addition to the 47 parameters by considering +-50% variations from their default values. For both small and large HH-model scenarios, we test total sample sizes of both 32k and 64k for all methods to evaluate their scalability with the number of simulated samples given during optimization. The results show that existing methods show good performances for small HH-model scenarios that scale with sample size consistent with literature. EP-GAN on the other hand shows overall better performance in predicting membrane potential responses on both small and large HH-model scenarios.

**Reviewer #2 (Public review):**
Major 1: Models do not faithfully capture empirical responses. While the models generated with EPGAN reproduce the average voltage during current injections reasonably well, the dynamics of the response are generally not well captured. For example, for the neuron labeled RIM (Figure 2), the most depolarized voltage traces show an initial 'overshoot' of depolarization, i.e. they depolarize strongly within the first few hundred milliseconds but then fall back to a less depolarized membrane potential. In contrast, the empirical recording shows no such overshoot. Similarly, for the neuron labeled AFD, all empirically recorded traces slowly ramp up over time. In contrast, the simulated traces are mostly flat. Furthermore, all empirical traces return to the pre-stimulus membrane potential, but many of the simulated voltage traces remain significantly depolarized, far outside of the ranges of empirically observed membrane potentials. The authors trained an additional GAN (EPGAN Extended) to improve the fit to the resting membrane potential. Interestingly, for one neuron (AWB), this improved the response during stimulation, which now reproduced the slowly raising membrane potentials observed empirically, however, the neuron still does not reliably return to its resting membrane potential. For the other two neurons, the authors report a decrease in accuracy in comparison to EP-GAN. While such deviations may appear small in the Root mean Square Error (RMSE), they likely indicate a large mismatch between the model and the electrophysiological properties of the biological neuron. The authors added a second metric during the revision - percentages of predicted membrane potential trajectories within empirical range. I appreciate this additional analysis. As the empirical ranges across neurons are far larger than the magnitude of dynamical properties of the response ('slow ramps', etc.), this metric doesn't seem to be well suited to quantify to which degree these dynamical properties are captured by the models.

We made improvements to the training data generation and architecture of EP-GAN to improve its overall accuracy with predicted membrane potential responses. In particular, we divided training data generation into three neuron types found in *C. elegans* non-spiking neurons: (1) Transient outward rectifier, (2) Outward rectifier and (3) Bistable [8, 16]. Each randomly generated training sample is categorized into one of 3 types by evaluating its steady-state currents with respect to experimental dI/dV bound constraints (See generating training data section under Methods for more detail). The process is then followed by imposing minimum-maximum constraints on simulated membrane potential responses. The setup allows generations of training samples that are of closer distribution to experimentally recorded neurons. This is further described in Section Methods page 15 in the revised manuscript.

We also improved the EP-GAN training process by incorporating random masking of input membrane potential responses. The masking forces EP-GAN to make predictions even with missing voltage traces, improving overall accuracy and allowing EP-GAN to use membrane potential inputs with arbitrary clamping protocol (see Methods page 13 for more detail). For the training loss functions, we further supplemented the membrane potential regression loss with errors computed for 2 intervals: pre- and post-stimulation time intervals to improve EP-GAN prediction capabilities for baseline membrane potentials.

Taken together, these modifications improved EP-GAN’s overall ability to better capture empirical membrane potential responses and we show the results in Figure 2 – 5, Table S2, S3.

Major 2: Comparison with other approaches is potentially misleading. Throughout the manuscript, the authors claim that their approach outperforms the other approaches tested. But compare the responses of the models in the present manuscript (neurons RIM, AFD, AIY) to the ones provided for the same neurons in Naudin et al. 2022 (https://doi.org/10.1371/journal. pone.0268380). Naudin et al. present models that seem to match empirical data far more accurately than any model presented in the current study. Naudin et al. achieved this using DEMO, an algorithm that in the present manuscript is consistently shown to be among the worst of all algorithms tested. I therefore strongly disagree with the authors claim that a "Comparison of EP-GAN with existing estimation methods shows EP-GAN advantage in the accuracy of estimated parameters". This may be true in the context of the benchmark performed in the study (i.e., a condition of very limited compute resources - 18 generations with a population size of 600, compare that to 2000 generations recommended in Naudin et al.), but while EP-GAN wins under these specific conditions (and yes, here the authors convincingly show that their EP-GAN produces by far the best results!), other approaches seem to win with respect to the quality of the models they can ultimately generate.

We thank the reviewer for the feedback regarding the comparison with existing methods. We have revised the optimization setup for existing methods (GDE3, NSDE, DEMO, and NSGA2) by incorporating steady-state response constraints as the initial selection process. The process is similar to that of EP-GAN training data generation and DEMO parameter selection process [16] (see Results section, page 6 for detail). Incorporating this process has improved the accuracy of existing methods especially for small HH-model scenarios where DEMO stood out with the best performance alongside NSGA2 (Figure 5, Table 1, 2).

We also expanded the testing scenarios by evaluating all methods with respect to both small and large HH-model estimation. The small HH-model scenario estimates 47 parameters consisting of channel conductance, reversal potentials and initial conditions with the channel parameters (n = 129) frozen to default values in [41]. Large HH-model includes estimating channel parameters (i.e. 129) in addition to the 47 parameters by considering +-50% variations from their default values. For both small and large HH-model scenarios, we test total sample sizes of both 32k and 64k for all methods to evaluate their scalability with the number of simulated samples given during optimization. The results show that existing methods show good performances for small HH-model scenarios that scale with sample size. EP-GAN on the other hand shows overall better performance in predicting membrane potential responses on both small and large HH-model scenarios.

In particular, with extended membrane potential error including pre-, mid- , post-activation periods, EP-GAN (trained with 32k samples, large HH-model, 9 neurons) mean membrane potential responses error of 2.82mV was lower than that of DEMO (12.2mV, 64k samples) trained on identical setup (Table 2) and DEMO (7.78mV, using 36,000k samples, 3 neurons) applied to simpler HHmodel in [16]. With respect to DEMO performance in [16], under identical simulation protocol (i.e., no stimulation during (0, 5s), (10, 15s) and stimulation during (5, 10s)), EP-GAN predicted RIM (large HH-model) showed membrane potential accuracy on par with that of DEMO (simpler HH-model) and EP-GAN predicted AFD showed better accuracy for post-activation membrane potential response where DEMO predicted membrane potentials overshoot above the baseline (not shown in the paper).

Major 3: As long as the quality of the models generated by the EP-GAN cannot be significantly improved, I am doubtful that it indeed can contribute to the 'ElectroPhysiome', as it seems likely that dynamics that are currently poorly captured, like slow ramps, or the ability of the neuron to return to its resting membrane potential, will critically affect network computations. If the authors want to motivate their study based on this very ambitious goal, they should illustrate that single neuron model generation with their approach is robust enough to warrant well-constrained network dynamics. Based on the currently presented results, I find the framing of the manuscript far too bold.

We thank the reviewer for the feedback regarding the paper's scope. With revised methods, the overall quality of EP-GAN models is improved with the most significant improvements in baseline membrane potential accuracy. While high quality neuron models could be attained with existing methods given sufficient sample size, our results suggest EP-GAN can predict models with enhanced quality with significantly fewer sample size without a need for retraining, thus complementing the main drawback of evolutionary based methods. While EP-GAN still has limitations (e.g., difficulty in predicting slow ramps) that need to be addressed in the future, we believe its overall performance combined with fast inference speed and flexibility in its input data format (e.g., missing membrane potential traces) is a step forward in the large-scale neuron modeling tasks that can contribute to network models.

Major 4: The conclusion of the ablation study 'In addition the architecture of EP-GAN permits inference of parameters even when partial membrane potential and steady-state currents profile are given as inputs' does not seem to be justified given the voltage traces shown in Figure 3. For example, for RIM, the resting membrane potential stays around 0 mV, but all empirical traces are around -40mV. For AFD, all simulated traces have a negative slope during the depolarizing stimuli, but a positive slope in all empirically observed traces. For AIY, the shape of hyperpolarized traces is off. While it may be that by their metric neurons in the 25% category are classified as 'preserving baseline accuracy', this doesn't seem justified given the voltage traces presented in the manuscript. It appears the metric is not strict enough.

We improved EP-GAN’s training process by incorporating random masking of input membrane potential responses. The masking forces EP-GAN to make predictions even with missing voltage traces, improving overall accuracy and allowing EP-GAN to use membrane potential inputs with arbitrary clamping protocol.

Such input masking during training has improved the results with ablation studies where EP-GAN now retains baseline membrane potential error (3.3mV, averaged across pre-, mid-, post-activation periods) up to 50% of membrane potential inputs remaining (3.5mV) and up to 25% of steady-state currents remaining (3.5mV).